# ProAct: A Benchmark and Multimodal Framework for Structure-Aware Proactive Response

Xiaomeng Zhu [1 2]   Fengming Zhu [1]   Weijie Zhou [2]   Ye Tian [2]   Zhenlin Hu [3]   Yufei Huang [2]   Yuchun Guo [2]
Xinyu Wu [4]   Zhengyou Zhang [2]   Fangzhen Lin [1]   Xuantang Xiong [2]

## Abstract

While passive agents merely follow instructions, proactive agents align with higher-level objectives, such as assistance and safety by continuously monitoring the environment to determine when and how to act. However, developing proactive agents is hindered by the lack of specialized resources. To address this, we introduce **ProAct-75**, a benchmark designed to train and evaluate proactive agents across diverse domains, including assistance, maintenance, and safety monitoring. Spanning 75 tasks, our dataset features 91,581 step-level annotations enriched with explicit task graphs. These graphs encode step dependencies and parallel execution possibilities, providing the structural grounding necessary for complex decision-making. Building on this benchmark, we propose **ProAct-Helper**, a reference baseline powered by a Multimodal Large Language Model (MLLM) that grounds decision-making in state detection, and leveraging task graphs to enable entropy-driven heuristic search for action selection, allowing agents to execute parallel threads independently rather than mirroring the human's next step. Extensive experiments demonstrate that ProAct-Helper outperforms strong closed-source models, improving trigger detection mF1 by 6.21%, saving 0.25 more steps in online one-step decision, and increasing the rate of parallel actions by 15.58%. Code is available at `https://github.com/ZhuXMMM/ProAct.git`

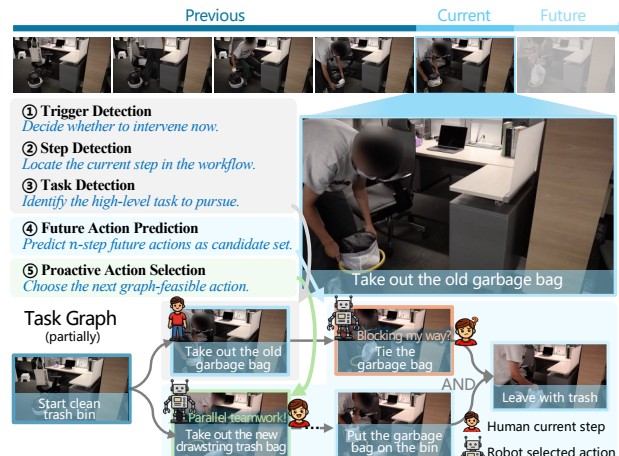

*Figure 1.* **Overview of proactive response tasks.** ProAct-75 supports five vision-based tasks with step-level annotations, hierarchical labels, and task graphs. Traditional intent-following approaches predict human-intended actions (*e.g.*, tie the bag) and execute them, inadvertently blocking workflows. Our benchmark enables evaluation of strategies where robots pursue independent parallel threads to reduce disruptions.

## 1. Introduction

Unlike passive agents that respond to explicit instructions, proactive agents take initiative toward higher-level objectives by continuously observing the environment, and autonomously selecting actions (Wooldridge & Jennings, 1995; Li et al., 2023; van Den Broek & Moeslund, 2024). For instance, a robot may replace a trash bag before it overflows in a human-absent scenario, or proactively prepare a new bag while observing a human remove a full one in a collaborative scenario. However, most existing robotic systems still rely on passive instructions, imposing cognitive load and limiting the robot's autonomous operation (Johannsmeier & Haddadin, 2016; Camilleri et al., 2022; Noormohammadi-Asl et al., 2025). In this work, we study proactive response for robot agents in settings of human-absent autonomy and human-robot collaboration, where agents must continuously monitor video observations to determine when to intervene and what action to take.

Training such proactive agents requires a robust ability of

[1] Department of Computer Science and Engineering, The Hong Kong University of Science and Technology (HKUST), Hong Kong SAR, China [2] Tencent, Shenzhen, China [3] Futian Laboratory, Shenzhen, China [4] Shenzhen Institute of Advanced Technology (SIAT), Chinese Academy of Sciences, Shenzhen, China. Correspondence to: Fangzhen Lin <flin@cse.ust.hk>, Xuantang Xiong <sheltxiong@tencent.com>.

*Proceedings of the $43^{rd}$ International Conference on Machine Learning*, Seoul, South Korea. PMLR 306, 2026. Copyright 2026 by the author(s).

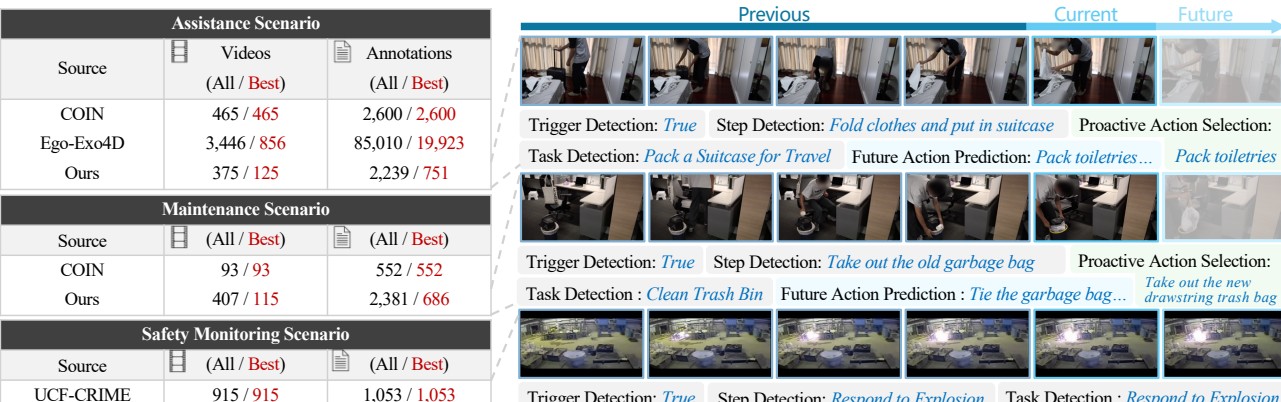

*Figure 2.* **Qualitative examples of ProAct-75 across three application scenarios.** We visualize the previous-current-future observation window and structured annotations for our proactive visual response tasks. Assistance and Maintenance examples are from self-collected exocentric videos. Safety examples are from UCF-Crime. Safety videos omit future action prediction and proactive action selection due to the absence of human-robot collaboration.

perception across diverse scenarios (Triantafyllidis et al., 2023; Gao et al., 2023; Wu et al., 2024). More importantly, they need to have structured task representations that connect high-level objectives with executable steps (Kaelbling & Lozano-Pérez, 2011; Kou et al., 2024; Wang et al., 2025b). An example is hierarchical task graphs with precedence constraints and parallel threads (Gombolay et al., 2018; Suslova & Fazli, 2020; Zhao et al., 2026; Kou et al., 2026) that enable structure-aware execution to preserve task feasibility and shorten workflows that enable robots to execute tasks that humans would eventually perform but have no precedence dependencies (Pupa et al., 2022). For instance, in the trash-handling scenario shown in Figure 1, a robot agent lacking dependency knowledge might wait to tie the bag sequentially, whereas understanding parallel threads would allow it to prepare a new bag concurrently, accelerating completion.

However, existing video understanding benchmarks present critical gaps for evaluating proactive response. While multi-source datasets are essential for cross-scenario diversity, they often exhibit inconsistent temporal granularities and annotation schemes (Sultani et al., 2018; Das et al., 2019; Tang et al., 2019; Damen et al., 2022; Zhu et al., 2023; Kou et al., 2023; Li et al., 2024; Hartmann et al.). Some annotate mid-level states while others focus on atomic actions, preventing unified hierarchical task modeling. More critically, existing benchmarks rarely provide task graphs that encode temporal precedence and parallel execution possibilities, which are essential for effective planning. Without explicit dependency structures, agents must treat all steps conservatively as sequential, missing opportunities to execute independent tasks in parallel and thereby unnecessarily prolonging workflows (Xiang et al., 2023; Zhu et al., 2025).

To address these gaps, we introduce ProAct-75, a benchmark for vision-based proactive response spanning **assistance, maintenance, and safety monitoring**. ProAct-75 comprises 75 tasks with 5,383 videos and 91,581 annotated segments. Videos are sourced from Ego-Exo4D (Grauman et al., 2024), COIN (Tang et al., 2019), and UCF-Crime (Das et al., 2019), complemented by 495 self-collected clips to improve coverage of underrepresented tasks. For evaluation, we adopt a consistent protocol to re-annotate all videos, ensuring atomic action-level temporal granularity. Critically, ProAct-75 provides explicit task graphs for each task, encoding AND/OR dependencies and parallelizable threads to support structure-aware action selection (enabling parallel support actions under constraints). This enables evaluation of five proactive response tasks: **trigger detection, step detection, task detection, future action prediction, and proactive action selection** (as shown in Figure 1), covering both intervention judgment and graph-feasible action selection beyond standard action recognition or anticipation.

Furthermore, we propose ProAct-Helper as a reference baseline built upon MLLM. ProAct-Helper employs a MLLM with a Hierarchical Binding Module (HBM) to enhance cross-level semantic consistency for perception tasks (trigger, task, step detection, and future action prediction)[1]. ProAct-Helper employs an entropy-driven heuristic to search for the next best proactive action on the given task graph, such that it may prioritize actions on parallel threads rather than strictly following the human's next intended step, and meanwhile, does not violate any precedence constraints. In summary, our main contributions are:

- A benchmark called ProAct-75 for vision-based proactive response that provides explicit task graphs and structure-aware annotations across assistance, mainte-

---

[1]We use "step" to refer to observed human activity states and "action" to refer to robot execution primitives, though both represent atomic nodes in the task graph.

nance, and safety monitoring scenarios.

- An MLLM-based framework called ProAct-Helper that integrates HBM for multi-level state perception and entropy-driven heuristic search for proactive action selection under task-graph constraints.
- Comprehensive experiments showing that ProAct-Helper outperforms strong MLLMs, improving trigger detection mF1 by 6.21%, achieving 0.25 saved steps, and increasing Parallel Action by 15.58%.

**Conflict of Interest Disclosure.** Some authors are affiliated with Tencent, which provided research support and data-collection resources for this work. The use of self-collected data followed the applicable company data-governance procedures. The authors independently conducted the benchmark design, annotation protocol, experiments, analysis, and conclusions. We are not aware of any other financial conflicts of interest.

## 2. Related Work

### 2.1. Proactive Embodied Agents

Existing proactive robotic systems typically aim to infer human goals from observations (Huang & Mutlu, 2016; Nikolaidis et al., 2017; Losey et al., 2018; Patel & Chernova, 2023). Related research adopts the inverse reasoning paradigm of Theory of Mind (ToM), inferring human intentions from behavioral observations to generate assistive strategies. These approaches often assume the robot agent's intentions align with human's intentions, positioning the robot as assistive tools (Jara-Ettinger et al., 2016; van Den Broek & Moeslund, 2024). Existing methods mainly cover intention recognition, environment prediction, and shared autonomy (Schrempf et al., 2005; Baker et al., 2009; Dragan et al., 2013; Koppula & Saxena, 2015; Javdani et al., 2018; Ognibene et al., 2019; Rhinehart et al., 2019; Broad et al., 2019; Shi et al., 2021; Atan et al., 2024; Wang et al., 2025a; 2026). However, recent studies indicate that robot agent's intentions need not align with human's intentions (van Den Broek & Moeslund, 2024; Zhu et al., 2025), as human intentions may be suboptimal in unattended scenarios or when holding negative beliefs (Wang et al., 2025c). This suggests robot agent should make more independent decisions based on scene understanding and task structure. While recent works (Bi et al., 2024; Yang et al., 2025; 2026) focus on sensory-driven user assistance, our work studies proactive response in attended and unattended settings, using task-graph structure to guide action selection from visual state estimates.

### 2.2. Video Benchmarks for Proactive Response

Existing proactive response research has primarily focused on text-based reasoning, such as ProRAC (Wu & Liu, 2025) for symbolic action reasoning and ProactiveBench (Wang et al., 2025d) for diagnostic evaluations. However, visual input is essential for proactive systems to perceive real-time environmental states and anticipate task requirements in physical scenarios. While large-scale egocentric datasets such as Ego4D (Grauman et al., 2022), EPIC-Kitchens (Damen et al., 2022), and Something-Something (Goyal et al., 2017) have advanced activity understanding, proactive systems rely more on holistic scene context and structured task representations from exocentric views (Patel & Chernova, 2023). Complete task-step sequences are particularly critical for learning procedural dependencies (Gao et al., 2022; Zhou et al., 2023). Existing exocentric datasets include COIN (Tang et al., 2019), Ego-Exo4D (Grauman et al., 2024), CrossTask (Zhukov et al., 2019), Assembly101 (Sener et al., 2022), and Toyota SmartHome (Das et al., 2019). We select COIN and Ego-Exo4D for their complete task-step annotations and multi-view collaboration data, introduce UCF-Crime (Sultani et al., 2018) for anomalous scenarios in safety monitoring, and collect additional videos for underrepresented tasks.

## 3. Problem Statement

Vision-based proactive response couples state perception with action selection under task-graph constraints. We formalize the problem in this section.

### 3.1. Task Graph Formulation

Human activities follow structured procedures: some steps must precede others, while certain sub-processes can progress in parallel. To provide an explicit executable constraint model for proactive response, we represent each task as a Directed Acyclic Graph (DAG) (Sifat et al., 2023; Grauman et al., 2024) with AND/OR dependencies. We provide the formal definitions below.

A *task T* is a finite directed graph $(V, E)$, denoting a finite set of nodes and the set of edges, respectively:

- A node $v \in V$ represents either an executable step ($v \in V_e$) or a non-executable structural node ($v \in V_n$). Non-executable nodes include start/terminate nodes and mid-level node pairs[2], with $V_e \cap V_n = \emptyset$.
- A directed edge $(u, v) \in E$ denotes a dependency where $u$ must be executed before $v$.
- The set of predecessor node is defined as $\mathrm{Pred}(v) \triangleq \{u \in V : (u, v) \in E\}$, and the successor node set as $\mathrm{Succ}(v) \triangleq \{w \in V : (v, w) \in E\}$.

---

[2]Mid-level nodes abstract over a category of behaviors or a sequence of actions, *e.g.*, "prepare ingredients". They come in pairs: a start node marks the beginning of a mid-level task and an end node marks its completion.

A node $a$ is *reachable* from node $b$, denoted $a \in \mathrm{Reach}(b)$, iff either $a = b$ (every node reaches itself), or there exists a directed path from $b$ to $a$. Formally, this is captured by the recursive definition:

$$a \in \mathrm{Reach}(b) \iff a = b \lor \exists c \in V. \, (c, a) \in E \land c \in \mathrm{Reach}(b). \tag{1}$$

Note that for any mid-level start node $u$ with end node $u'$ and successors $b_i, b_j \in \mathrm{Succ}(u)$, there exists no edge $(v_i, v_j) \in E$ or $(v_j, v_i) \in E$ where $v_i \in (\mathrm{Reach}(b_i) \cap \mathrm{Pred}(u')) \setminus \mathrm{Reach}(b_j)$ and $v_j \in (\mathrm{Reach}(b_j) \cap \mathrm{Pred}(u')) \setminus \mathrm{Reach}(b_i)$, ensuring branch independence until merge to $u'$.

We associate each node with type via $\phi : V \mapsto \{\mathrm{AND}, \mathrm{OR}\}$, which specifies when the node can be executed:

- For a node $v$ with $\phi(v) = \mathrm{AND}$, it can only be executed when every node in $\mathrm{Pred}(v)$ has been executed. We call it an AND node for simplicity.
- Similarly, for a node $v$ with $\phi(v) = \mathrm{OR}$, it can only be *executed* when at least one node in $\mathrm{Pred}(v)$ has been executed, and is termed an OR node similarly.
- In particular, for the initial step $v_0$ with $\mathrm{Pred}(v_0) = \emptyset$, it can be executed immediately.

Let $\mathrm{Prog}_t \subseteq V$ denote the set of executed nodes up to timestep $t$. Every executable step $v \in V_e$ consumes a timestep upon execution, while a structural node $v \in V_n$ is automatically satisfied once its preconditions are met. Task progression follows the constraints:

$$\forall v \in \mathrm{Prog}_t. \begin{cases} \phi(v) = \mathrm{AND} \implies \mathrm{Pred}(v) \subseteq \mathrm{Prog}_t \\ \phi(v) = \mathrm{OR} \implies \mathrm{Pred}(v) \cap \mathrm{Prog}_t \neq \emptyset \end{cases} \tag{2}$$

An executable step $a \in V_e$ is legal at timestep $t$ if $a \notin \mathrm{Prog}_t$ and its preconditions are satisfied under the AND/OR semantics above. We denote the set of such steps by $\mathcal{A}_t^{\mathrm{legal}} \subseteq V_e$. A task is completed at timestep $t^*$ if the current progression state reaches the terminal node $v_{\mathrm{term}}$ for the first time, *i.e.*, $v_{\mathrm{term}} \in \mathrm{Prog}_{t^*} \land \forall t' < t^*. v_{\mathrm{term}} \notin \mathrm{Prog}_{t'}$.

### 3.2. Proactive Response Formulation

Proactive response requires continuously monitoring human activities and intervening when assistance is needed. At each timestep $t$, given a video frame and task graph as the input $\mathbf{X}_t$, the agent outputs $\mathbf{Y}_t = (y_t^{\mathrm{trig}}, y_t^{\mathrm{task}}, y_t^{\mathrm{step}}, \hat{a}_{t+1:t+n}, a_{t+1}^{\star})$:

- **Trigger**: $y_t^{\mathrm{trig}} \in \{0, 1\}$ indicates if interaction is needed.
- **Task**: $y_t^{\mathrm{task}} \in \mathcal{T}$ identifies the task category.
- **Step**: $y_t^{\mathrm{step}} \in V$ identifies the current step.
- **Future actions**: $\hat{a}_{t+1:t+n} \in V$ predicts future actions.
- **Proactive action**: $a_{t+1}^{\star} \in V_e \cup \{\textsc{Wait}\}$ is the robot's selected next action.

These predictions form a hierarchical structure where trig-

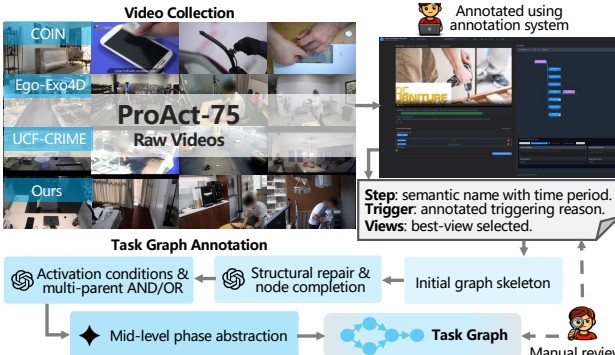

*Figure 3.* **ProAct-75 data collection and annotation pipeline.** We combine videos from public datasets and self-collected recordings, then annotate step spans/names, triggers, and best views. Each task is equipped with a task-graph annotation.

ger gates subsequent predictions, and task/step detection localizes the current state for action planning.

## 4. ProAct-75 Benchmark

ProAct-75 evaluates proactive response on trigger detection, task detection, step detection, future action prediction, and proactive action selection tasks. In the following sections we describe the data composition and annotation protocol.

### 4.1. Data Composition and Annotations

ProAct-75 covers three interaction scenarios with distinct mechanisms of goal formation. **Assistance** evaluates anticipatory support in activities with human-initiated goals. **Maintenance** evaluates environment-triggered goal generation driven by observable states (*e.g.*, cluttered desks). **Safety monitoring** focuses on preventive interventions against risky behaviors. Assistance and Maintenance may overlap at the task level depending on whether intervention is triggered by human activity or environment monitoring.

To cover these scenarios at scale while maintaining diverse environments, we construct ProAct-75 by aggregating exocentric videos from Ego-Exo4D, COIN, UCF-Crime, and self-collected sources. We adopt the Ego-Exo4D standard and re-annotate other sources to ensure consistent step-level granularity. For multi-view recordings in Ego-Exo4D, we follow the annotation of original view. For our self-collected multi-view videos, we select the best views so that key actions are visible and occlusions are minimized. The data scale and distribution are shown in Figure 2. More collection details are available in Section D.

We adopt a unified annotation protocol where each step boundary corresponds to the semantic change in human actions. Each video is annotated with task- and step-level temporal spans (aligned timestamps and frame indices), and natural-language context cues (scenario and trigger descrip-

tions at task and step granularity). We provide human-rated task priority scores to support cost-sensitive evaluation. For COIN, UCF-Crime, and our selected data, three trained annotators produced 1,978 videos and 6,797 step segments over 1.5 months. Quality control involved two rounds (half a month) where three experts reviewed annotations to fix inconsistencies and refine boundaries. Disagreements were resolved through deliberation to ensure consistency across sources. In total, ProAct-75 comprises 5,383 videos containing 91,581 step segments, with the remainder contributed by Ego-Exo4D's existing step annotations.

### 4.2. Task Graph

Following the formulation in Section 3.1, we construct task graphs as DAGs with atomic steps as nodes and temporal dependencies as edges. We build each graph incrementally. Given the step inventory of a task, GPT-4o proposes one local dependency (with activation conditions) at a time, followed by automatic validation to enforce DAG validity (acyclicity, reachability, no dead ends/isolated nodes) and basic physical feasibility checks. Gemini-3-Pro further groups atomic steps into mid-level phases. Finally, all graphs are manually reviewed to correct any residual inconsistencies (as shown in Figure 3). More statistics on the task graphs can be found in Section E.

## 5. ProAct-Helper Method

### 5.1. ProAct-Helper Framework

ProAct-Helper is a proactive response framework built upon MLLM and integrated with task-graph planning. The model takes multimodal input $\mathbf{X}_t$ at time $t$ and produces structured predictions $\mathbf{Y}_t = (y_t^{\text{trig}}, y_t^{\text{task}}, y_t^{\text{step}}, \hat{a}_{t:t+\tau}, a_{t+1}^{\star})$ as defined in Section 3.2. To handle the long-tailed trigger–task–step hierarchy, we propose HBM to enforce cross-level alignment and improve rare-class representations.

Our training process employs instruction tuning with auxiliary constraints. We apply standard autoregressive cross-entropy loss $L_{\text{CE}}$ over supervised tokens, supplemented by a binary classification loss $L_{\text{trig}}$ at the trigger token position to prevent signal dilution. The final objective is:

$$L = L_{\text{CE}} + \lambda_{\text{trig}} L_{\text{trig}} + \lambda_{\text{bind}} L_{\text{bind}}, \tag{3}$$

where $L_{\text{bind}}$ is introduced by the Hierarchical Binding Module to strengthen cross-level consistency and mitigate training instability for long-tail classes.

### 5.2. Hierarchical Binding Module

The supervision signals in ProAct-75 exhibit a trigger-task-step hierarchy with severe long-tail distributions at the task and step levels. To address the representation insufficiency caused by relying solely on autoregressive supervi-

sion, HBM enhances cross-level semantic consistency by maximizing the agreement between parent and child level representations while strengthening class separability under parent-level conditioning.

HBM extracts token hidden states for trigger/task/step fields (Figure 4). Let $h_i$ be the $i$-th output-token hidden state and $I_\ell$ the token span of level $\ell \in \{\text{trig}, \text{task}, \text{step}\}$. We obtain $H_\ell$ by mean pooling over $\{h_i\}_{i \in I_\ell}$. We introduce two cross-level contrastive constraints to maximize mutual information between hierarchical levels while respecting category structure. For each paired instance $i$ in a mini-batch of size $B$ with parent-child pair $(H_p^{(i)}, H_c^{(i)})$, we define the positive pair as $(H_p^{(i)}, H_c^{(i)})$ and the negatives as mismatched pairs formed with other instances in the mini-batch. The binding loss between parent level $p$ and child level $c$ is:

$$\mathcal{L}(p, c) = -\frac{1}{B} \sum_{i=1}^{B} \log \frac{\exp(\text{sim}(H_p^{(i)}, H_c^{(i)})/\tau)}{\sum_{j \in \mathcal{N}} \exp(\text{sim}(H_p^{(i)}, H_c^{(j)})/\tau)}, \tag{4}$$

where $\mathcal{N} = \{1, \ldots, B\}$ indexes paired instances in the mini-batch, $\text{sim}(\cdot, \cdot)$ is cosine similarity, and $\tau$ is temperature. We use a symmetric variant by averaging $\mathcal{L}(p, c)$ and $\mathcal{L}(c, p)$.

We combine two cross-level binding terms as:

$$L_{\text{bind}} = \lambda_{tt} L_{\text{trig2task}} + \lambda_{ts} L_{\text{task2step}}. \tag{5}$$

where $L_{\text{trig2task}}$ and $L_{\text{task2step}}$ are instantiated as $\mathcal{L}(\text{trig}, \text{task})$ and $\mathcal{L}(\text{task}, \text{step})$, respectively.

### 5.3. Proactive Action Selection

In this work, we study proactive action selection using annotated task graphs to choose the next feasible step. The key challenge is exploiting parallelizable thread to enable concurrent progress, rather than strictly following a single sequential path. Inspired by the perspective of behavioral entropy (Goodrich et al., 2004; Guastello et al., 2012; Balch, 2000), we model the human/robot action distribution over parallel threads and use entropy to penalize mixed thread assignments, which indicate frequent thread switching and higher cognitive load.

To capture parallelizable threads in collaborative tasks, we define threads based on the reachability structure of the task graph. For any mid-level start node $u$ with multiple successors $\text{Succ}(u) = \{b_1, \ldots, b_m\}$ that later merge at $u'$, we assign each successor (and its downstream nodes) to a thread via a mapping $\pi : V \mapsto \mathbb{N}$. Two successors $b_i$ and $b_j$ are considered in the same thread if their reachable node sets overlap, i.e., $\text{Reach}(b_i) \cap \text{Reach}(b_j) \neq \emptyset$; otherwise they belong to different threads. Nodes not covered by such regions are assigned to the primary thread $\pi(v) = 0$.

Given the task graph, current progression state, and action set $\mathcal{A}$, we obtain the set of legal actions $\mathcal{A}_t^{\text{legal}}$. The thread

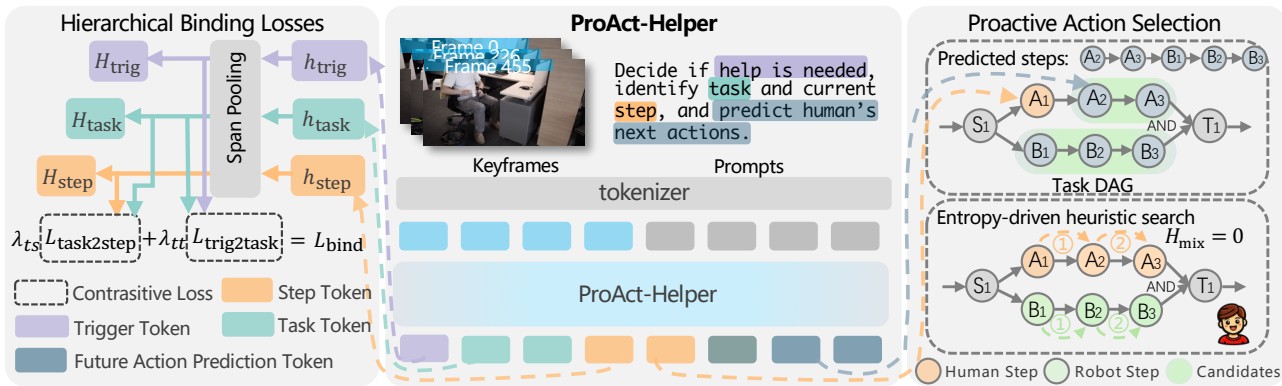

*Figure 4.* **Overview of ProAct-Helper framework.** Given keyframes and prompts, the model predicts trigger, task, step, and human's future actions, trained with hierarchical binding losses for cross-level consistency under long-tail data. It then selects the next robot action on the task DAG via an entropy-driven heuristic search to favor feasible, low thread-mixing progress.

mapping $\pi(\cdot)$ groups these legal actions by thread, and the robot agent selects an action from the grouped set based on the objective below. For each action $a \in \mathcal{A}_t^{\text{legal}}$, we estimate its induced thread mixing entropy $H_{\text{mix}}$.

Let $\mathcal{H}_t$ and $\mathcal{R}_t$ denote the human and robot step histories up to time $t$. To evaluate the thread mixing induced by a candidate action $a$, we consider the counterfactual one-step-ahead robot history $\mathcal{R}_t \cup \{a\}$ and compute the per-thread step count for each agent $\alpha \in \{\text{hum}, \text{rob}\}$ as:

$$n_k^\alpha \triangleq \begin{cases} |\{x \in \mathcal{H}_t \mid \pi(x) = k\}| & \text{if } \alpha = \text{hum}, \\ |\{x \in \mathcal{R}_t \cup \{a\} \mid \pi(x) = k\}| & \text{if } \alpha = \text{rob}. \end{cases} \quad (6)$$

The mixing ratio for thread $k$ is then $p_k = n_k^{\text{hum}}/(n_k^{\text{hum}} + n_k^{\text{rob}})$, quantifying the proportion of human participation. The binary entropy for each thread is:

$$H_k(p_k) = -p_k \log p_k - (1 - p_k) \log(1 - p_k), \quad (7)$$

where $H_k(p_k) = 0$ when $p_k \in \{0, 1\}$ (*i.e.*, single-agent execution). The thread mixing entropy is a length-weighted sum across threads:

$$H_{\text{mix}}(\mathcal{H}_t, \mathcal{R}_t \cup \{a\}) = \sum_k w_k H_k(p_k), \quad (8)$$

where $w_k = (n_k^{\text{hum}} + n_k^{\text{rob}})/\sum_j(n_j^{\text{hum}} + n_j^{\text{rob}})$ weights threads by total executions, so mixing on more active threads contributes more than on rarely visited ones.

We employ a one-step lookahead strategy to select the action that minimizes this entropy:

$$a_{t+1}^\star = \arg\min_{a \in \mathcal{A}_t^{\text{legal}}} H_{\text{mix}}(\mathcal{H}_t, \mathcal{R}_t \cup \{a\}). \quad (9)$$

This strategy favors actions on threads different from the human's current thread and discourages frequent switching by the robot, reducing coordination overhead while enabling parallel progress.

## 6. Experiments

### 6.1. Experimental Setup

**Benchmark.** All experiments use ProAct-75, split by video into train/test at an approximate 3:1 ratio, yielding $N_{\text{train}} = 4,074$ and $N_{\text{test}} = 1,309$ videos. Unless stated otherwise, we train on the best-view subset ($N_{\text{train}} = 1,905$ and $N_{\text{test}} = 516$; see Section 4.1 and Figure 2) and evaluate on its test set, while other views are used for Out Of Distribution (OOD) evaluation (see Section B.5).

**Implementation details.** We extract keyframes based on adjacent-frame appearance changes and use a 5-frame sliding window with stride 3 as the input $\mathbf{X}_t$ (see Section B.1). Input prompt templates and output formats are detailed in Section F. For action selection, since predicted actions may not fall within the legal action domain, we take the intersection $\mathcal{A}_t^{\text{cand}} = \mathcal{A}_t^{\text{legal}} \cap \mathcal{A}_t^{\text{pred}}$ to ensure feasibility. We slightly abuse the notation by treating $\mathcal{A}_t^{\text{pred}}$ as a set and removing possible duplicates.

We evaluate both open-source and closed-source MLLMs. ProAct-Helper is built upon Qwen2.5-VL-Instruct 3B/7B and fine-tuned with LoRA. All baselines are evaluated in two stages: Stage-1 decides trigger and task given the keyframe window and task list; if triggered, Stage-2 detects step and future actions given the identified task graph. We avoid in-context video demonstrations as keyframe streams already create long video-token contexts, and adding demo keyframes would substantially increase latency.

ProAct-Helper is trained for 10 epochs with a batch size of 128 using the AdamW optimizer and a learning rate of $5 \times 10^{-5}$. All training is conducted on NVIDIA H20 GPUs using bfloat16 precision. For LoRA fine-tuning, we set the rank to $r = 32$ and the scaling factor to $\alpha = 32$. For inference, we apply different decoding strategies depending on the evaluation setup. When measuring generation time

on our fine-tuned model, we use greedy decoding with a maximum of 256 tokens. For the evaluation of baseline models, we employ nucleus sampling with temperature $T = 0.7$, top-$p = 0.8$, and top-$k = 20$.

**Evaluation Metrics.** For *trigger, task, and step detection*, we report **Acc/F1** as micro-averaged metrics and **mAcc/mF1**, which average over classes to reflect long-tail performance. For future action prediction, we compute the **ED** between predicted and ground-truth sequence. Furthermore, we report four metrics for *proactive action selection*. **Saved Steps (SS)** measures how many human steps are saved by robot actions; **Entropy (E)** quantifies the human-robot thread mixing entropy $H_{\text{mix}}$; **ER** measures thread entropy for robot threads; **Parallel Actions (PA)** measures the proportion of parallel actions where robot and previous human threads differ. E, ER, and PA are computed over effective actions only, *i.e.*, excluding wait.

## 6.2. Main Results

Table 1 presents results on ProAct-75. For all MLLMs baselines, evaluation follows a two-stage process. The first stage determines whether a trigger occurs and identifies the corresponding task based on the keyframe window and the task list. If triggered, the second stage uses the task graph of the identified task to detect current steps and predict future steps. For proactive action selection, which relies on textual predictions from the previous stage, we provide task graphs, AND/OR constraints, and thread definitions with parallel execution prioritization (as detailed in Section F).

The results show that ProAct-Helper outperforms both open-source and closed-source baselines across most metrics, validating its effectiveness for proactive assistance. *For trigger/task/step detection, ProAct-Helper (7B) improves task F1 by 17.09% and step F1 by 11.72% over the best closed-source baseline Gemini-2.5-Pro.* Furthermore, our HBM module boosts task mF1 by 2.71% and step mF1 by 1.63% on average across both backbones compared to the plain variant. This improvement on mF1 averaged over all the classes indicates better handling of long-tail categories.

For proactive action selection, ProAct-Helper significantly outperforms existing strong baseline models. *Compared to the best closed-source model Gemini-2.5-Pro, the 7B-based ProAct-Helper achieves SS of 0.361 and PA of 19.41%*, demonstrating stronger task parallelization capability. The plain variant without HBM attains lower SS at 0.350 and PA at 18.72%, as improved step detection and action prediction accuracy from HBM enable more effective action selection. Additional qualitative, hyperparameter, OOD, trigger-error, representation, and inference time analyses are in Section B.1 and Section C.

## 6.3. Ablation Study

To enable ablation studies within computational budgets, we sampled $1/8$ of the best-view split using stratified sampling, preserving task diversity and step-level distributions.[3] We examine HBM by isolating the Trigger-Task alignment loss $L_{\text{trig2task}}$ and Task-Step dependency loss $L_{\text{task2step}}$. Table 2 shows that introducing $L_{\text{trig2task}}$ improves Task and Step mAcc by 2.82% and 2.56%, while $L_{\text{task2step}}$ yields larger gains of 3.48% and 2.70%. The full model achieves best performance, and $L_{\text{task2step}}$ is consistently more effective than $L_{\text{trig2task}}$, indicating that grounding tasks in execution steps yields stronger supervision.

## 6.4. Analysis of Text-only Proactive Action Selection

To decouple proactive action selection from upstream action prediction errors, we simulate collaboration between human and robot agents from the video's initial state. In the oracle setting, the robot uses the complete human trajectory as $\mathcal{A}_t^{\text{pred}}$. For LLM-based methods, we adopt the prompt settings from Section 6.2. To avoid rollout deadlocks caused by minor mismatches between human traces and strict graph preconditions, we use a one-step alignment safeguard that temporarily admits the observed next human step when it is not graph-enabled. This is applied uniformly across methods and only at the current timestep, while robot actions remain graph-filtered. The full simulation procedure, including the safeguard and tie-breaking rule, is provided in Section A.

Table 3 shows that closed-source LLMs' SS remain lower than ProAct-Helper, despite being provided with task graphs and AND/OR constraints. GPT-4o exhibits lower mixing entropy $E$ than ProAct-Helper yet higher robot thread entropy ER. This suggests closed-source LLMs output more waiting actions, leaving more effective actions to be completed by the human alone, thereby reducing E while failing to establish stable robot parallel execution threads as evidenced by elevated ER. In contrast, ProAct-Helper achieves the lowest ER of 0.654 and highest PA of 33.95%, demonstrating that entropy-driven heuristic search more effectively drives strategies for stable parallel execution.

## 6.5. Analysis of Failure Case

Figure 5a reports hallucination rates, *i.e.*, outputs that violate predefined constraints. We consider three types: Trigger (false interventions when the ground truth requires none), Step (predicting step labels outside the predefined step vocabulary), and Future (empty sequences or steps outside the step vocabulary). ProAct-Helper achieves low Trigger and near-zero Step hallucination, indicating strong vocab-

---

[3]Stratified sampling ensures at least one video per task. Long-tail distributions arise intrinsically from task structures

*Table 1.* **Main results on ProAct-75.** We report mAcc/mF1 and Acc/F1 metrics for trigger/task/step detection and Edit Distance (ED) for future action prediction. Open-source baselines are evaluated without fine-tuning, while closed-source baselines use the same prompt template and decoding settings. Best results are in **bold** and second-best results are underlined. ↑ / ↓ indicate higher/lower is better.

| Base Model | Trigger (%) | | | | Task (%) | | | | Step (%) | | | | ED↓ | SS↑ | PA (%)↑ |
|---|---|---|---|---|---|---|---|---|---|---|---|---|---|---|---|
| | mAcc | mF1 | Acc | F1 | mAcc | mF1 | Acc | F1 | mAcc | mF1 | Acc | F1 | | | |
| *Open-Source MLLMs* | | | | | | | | | | | | | | | |
| Qwen3-VL-30B-A3B-Instruct | 51.61 | 66.25 | 73.50 | 81.89 | 22.80 | 31.70 | 34.66 | 51.48 | 6.67 | 9.44 | 8.53 | 15.72 | 4.77 | 0.013 | 0.09 |
| Qwen3-Omni-30B | 46.86 | 63.35 | 65.07 | 71.30 | 18.96 | 26.87 | 17.55 | 29.87 | 3.72 | 5.42 | 4.48 | 8.58 | 4.64 | 0.078 | 6.09 |
| Qwen2.5-VL-32B-Instruct | 45.72 | 62.36 | 63.77 | 69.64 | 16.62 | 23.96 | 29.45 | 45.50 | 4.04 | 6.04 | 6.14 | 11.57 | 4.72 | 0.038 | 3.71 |
| Qwen2.5-VL-3B-Instruct | 42.33 | 54.94 | 69.55 | 80.59 | 12.53 | 18.92 | 23.18 | 37.64 | 1.46 | 2.42 | 3.06 | 5.95 | 4.74 | 0.034 | 4.23 |
| *Closed-Source MLLMs* | | | | | | | | | | | | | | | |
| Qwen3-VL-Flash | 54.73 | 70.35 | 72.06 | 77.48 | 17.36 | 24.84 | 17.91 | 30.37 | 3.70 | 5.51 | 2.94 | 5.72 | 4.76 | 0.046 | 2.46 |
| Qwen3-VL-Plus | 51.52 | 67.39 | 69.89 | 76.41 | 14.88 | 21.52 | 17.39 | 29.63 | 5.14 | 7.38 | 4.66 | 8.91 | 4.71 | 0.045 | 3.10 |
| GPT-4o | 30.59 | 46.70 | 47.11 | 51.38 | 16.67 | 24.90 | 20.27 | 33.71 | 4.02 | 6.05 | 8.70 | 16.0 | 4.64 | 0.026 | 0.77 |
| Gemini-2.5-Flash | 49.73 | 64.24 | 72.52 | 81.44 | 21.78 | 30.66 | 31.14 | 47.49 | 6.97 | 10.12 | 9.42 | 17.21 | 4.70 | 0.062 | 2.55 |
| Gemini-2.5-Pro | 42.89 | 55.94 | 69.31 | 80.21 | 22.95 | 32.39 | 35.23 | 52.11 | **8.69** | **12.31** | 11.99 | 21.41 | 4.70 | 0.120 | 3.83 |
| ProAct-Helper (plain) | 61.50 | 75.40 | 79.24 | 85.12 | 24.02 | 31.85 | 48.21 | 65.05 | 6.46 | 9.27 | 17.54 | 29.85 | 3.99 | 0.333 | 17.15 |
| ProAct-Helper | 61.50 | 75.38 | 79.32 | 85.23 | **28.33** | **36.72** | 51.03 | 67.58 | 7.61 | 10.67 | 18.68 | 31.49 | 3.96 | **0.366** | 17.44 |
| *ProAct-Helper (based on Qwen2.5-VL-7B-Instruct)* | | | | | | | | | | | | | | | |
| ProAct-Helper (plain) | 62.36 | 76.09 | 79.85 | 85.57 | 25.49 | 34.24 | 48.94 | 65.71 | 6.39 | 9.70 | 18.23 | 30.83 | 3.99 | 0.350 | 18.72 |
| ProAct-Helper | **62.90** | **76.56** | **80.08** | **85.65** | 27.07 | 34.79 | 52.91 | 69.20 | 8.25 | 11.56 | **19.85** | **33.13** | **3.90** | 0.361 | **19.41** |

*Table 2.* **Ablation study results.** We report mAcc/mF1 and Acc/F1 for trigger/task/step detection and ED for future action prediction.

| Task →Step | Trigger →Task | Task (%) | | | | Step (%) | | | | ED↓ |
|---|---|---|---|---|---|---|---|---|---|---|
| | | mAcc | mF1 | Acc | F1 | mAcc | mF1 | Acc | F1 | |
| × | × | 13.43 (+0.00) | 18.50 (+0.00) | 33.22 (+0.00) | 49.87 (+0.00) | 2.64 (+0.00) | 3.99 (+0.00) | 9.26 (+0.00) | 16.94 (+0.00) | 4.23 (+0.00) |
| × | ✓ | 16.25 (+2.82) | 22.10 (+3.60) | 35.72 (+2.50) | 52.64 (+2.77) | 5.20 (+2.56) | 7.14 (+3.15) | 13.38 (+4.12) | 23.60 (+6.66) | 3.97 (-0.26) |
| ✓ | × | 16.91 (+3.48) | 22.60 (+4.10) | 39.08 (+5.86) | 56.19 (+6.32) | 5.34 (+2.70) | 7.28 (+3.29) | 13.70 (+4.44) | 24.10 (+7.16) | 4.01 (-0.22) |
| ✓ | ✓ | 17.12 (+3.69) | 23.10 (+4.60) | 42.06 (+8.84) | 59.22 (+9.35) | 5.25 (+2.61) | 7.18 (+3.19) | 13.96 (+4.70) | 24.50 (+7.56) | 3.94 (-0.29) |

*Table 3.* **Trajectory simulation results with human and robot agents under GT labels.**

| Method | SS↑ | E↓ | ER↓ | PA (%)↑ |
|---|---|---|---|---|
| *Closed-Source Large Language Models (LLMs)* | | | | |
| GPT-4o | 5.872 | **0.640** | 0.736 | 33.60 |
| Gemini-2.5-Flash | 5.918 | 0.662 | 0.748 | 28.26 |
| Gemini-2.5-Pro | 6.023 | 0.671 | 0.723 | 29.52 |
| DeepSeek-v3.2 | 5.868 | 0.664 | 0.742 | 31.58 |
| Qwen3-Max | 6.160 | 0.683 | 0.769 | 29.89 |
| *ProAct Action Selection Strategies* | | | | |
| Greedy | **9.868** | 0.837 | 0.836 | 28.11 |
| ProAct-Helper | **9.868** | 0.662 | **0.654** | **33.95** |

ulary alignment, but exhibits some Future hallucination, suggesting controllability issues in action sequence generation. Among closed-source models, GPT-4o shows the lowest overall hallucination, Gemini-2.5-Pro exhibits the highest Trigger hallucination, and Qwen3-VL variants show the highest Step and Future hallucination.

Figure 5b decomposes waiting into model-generated wait and forced wait (*i.e.*, caused by illegal actions). Gemini-2.5-Flash exhibits the lowest forced wait ratio, indicating better constraint adherence, while GPT-4o shows the highest, revealing reasoning limitations. Our method achieves the highest parallel action ratio, substantially exceeding all closed-source models, demonstrating a preference for parallel thread execution and efficient collaboration. Moreover, most closed-source models achieve parallel action ratios below Greedy, reflecting insufficient preference for parallel thread execution with humans under common-sense-driven decision making, thereby hindering efficient collaboration.

## 7. Conclusion

This paper studies proactive response, where an agent recognizes state from video and selects feasible next actions under task-graph constraints. To support this problem, we introduce **ProAct-75**, aggregating multi-source videos into step-level annotations with explicit task graphs across *assistance, maintenance, and safety scenarios*. On top of this

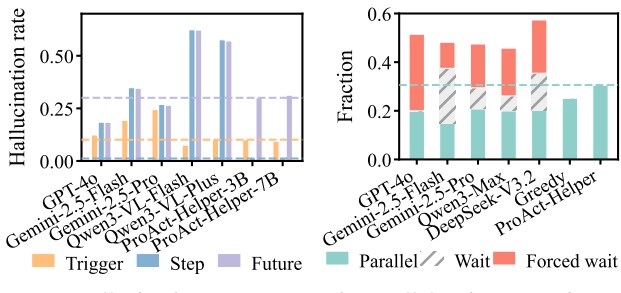

*(a)* Hallucination rates.     *(b)* Parallel action execution.

*Figure 5.* **Failure case analysis.** (a) We compare hallucination rates across trigger, step, and future prediction actions for different MLLMs. (b) We analyze models' parallel execution tendencies, waiting behaviors, and task graph constraint comprehension. For clarity, we omit non-parallel actions from the visualization.

benchmark, we find existing MLLMs often struggle to follow task-graph structure and leverage parallel threads even when the graphs are given as input. We therefore propose a strong multimodal baseline, **ProAct-Helper**. Extensive experiments show clear gains over closed-source models in state recognition and collaboration efficiency. Although our planner is a lightweight graph-constrained heuristic, its thread-entropy objective and DAG feasibility signals suggest future directions for graph-feasible decoding, and ProAct-75's trigger descriptions provide additional post-training data for reasonable proactive responses.

## Acknowledgments

We thank Changwei Wang and Weiheng Chi for helpful discussions and valuable feedback that improved this work.

## Impact Statement

This paper presents work whose goal is to advance the field of Machine Learning. Our benchmark is primarily built on publicly available datasets. We will release the corresponding annotations and evaluation code, and provide instructions for obtaining the original videos through the official channels of each source dataset. This avoids redistributing restricted content and ensures compliance with the respective licenses. For our self-collected videos, the data were collected under an industrial data-collection protocol, and all participants provided written informed consent for research and model-training use. The data collection and planned public release follow the applicable consent terms, internal compliance review, and company data-governance procedures. To mitigate privacy risks, we blur potentially sensitive information such as computer screens and logos, and anonymize participants' faces before release. A potential risk is unnecessary intervention caused by false-positive triggers, especially in safety-monitoring scenarios. ProAct-75 is intended as a benchmark for perception and decision

research rather than a ready-to-deploy autonomous safety system. High-stakes deployments should use calibrated trigger thresholds and human confirmation before consequential interventions.

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

## A. Implementation Details of Proactive Action Selection

We detail the evaluation procedure for proactive action selection under task-graph constraints. While Section 6.4 summarizes the rollout safeguard used in the full-video simulation, this appendix provides the complete procedure, including candidate construction, the deadlock-prevention safeguard, tie-breaking, and the SS/PA metrics. At each timestep $t$, the robot first constructs the legal action set $\mathcal{A}_t^{\text{legal}} \subseteq V_e$ according to the annotated AND/OR preconditions, then filters it by the predicted action sequence $\mathcal{A}_t^{\text{pred}}$ to obtain $\mathcal{A}_t^{\text{cand}}$ (Algorithm 1).

For the full-video *text-only* proactive action selection simulation in Table 3, we apply a deadlock-prevention safeguard. Human execution may not strictly satisfy task-graph preconditions (*e.g.*, minor omissions or implicit prerequisites), while our simulator enforces strict graph feasibility, which can stall rollouts when some nodes never become enabled. To maintain simulation fidelity, we use a one-step runtime alignment that relaxes feasibility *only* for the observed next human action. Let $g_{t+1}$ denote the ground-truth next human executable step at time $t$. If $g_{t+1} \in V_e$, $g_{t+1} \notin \text{Prog}_t$, and $g_{t+1} \notin \mathcal{A}_t^{\text{legal}}$, we temporarily augment $\widetilde{\mathcal{A}}_t^{\text{legal}} \leftarrow \mathcal{A}_t^{\text{legal}} \cup \{g_{t+1}\}$ before constructing $\mathcal{A}_t^{\text{cand}}$. This safeguard is applied uniformly across methods and only admits $g_{t+1}$ at the current timestep. Note that it is used only for Table 3, since Table 1 evaluates one-step decisions without full-rollout deadlocks.

Moreover, when multiple candidates attain the same minimum entropy value, we select the action that appears earliest in $\mathcal{A}_t^{\text{pred}}$ to ensure reproducibility. Concretely, $\text{Pos}(\mathcal{S}, a)$ returns the smallest index of $a$ in a sequence $\mathcal{S}$, and we apply a lexicographic $\arg\min$ over $(H_{\text{mix}}, \text{Pos})$.

---

**Algorithm 1** Thread-Entropy-Based Proactive Action Selection with Rollout Safeguard

---

**Require:** Task graph $G = (V, E)$, executable set $V_e$, thread mapping $\pi(\cdot)$, progression state $\text{Prog}_t$, ground-truth next human step $g_{t+1}$, histories $\mathcal{H}_t, \mathcal{R}_t$, predicted action sequence $\mathcal{A}_t^{\text{pred}}$

**Ensure:** Robot next action $a_{t+1}^\star$

    $\mathcal{A}_t^{\text{legal}} \leftarrow \{a \in V_e \mid a \notin \text{Prog}_t, \text{Precond}(a; \text{Prog}_t) = \text{true}\}$

    $\widetilde{\mathcal{A}}_t^{\text{legal}} \leftarrow \mathcal{A}_t^{\text{legal}}$

    **if** $g_{t+1} \in V_e$ **and** $g_{t+1} \notin \text{Prog}_t$ **and** $g_{t+1} \notin \mathcal{A}_t^{\text{legal}}$ **then**

        $\widetilde{\mathcal{A}}_t^{\text{legal}} \leftarrow \widetilde{\mathcal{A}}_t^{\text{legal}} \cup \{g_{t+1}\}$

    **end if**

    **if** $\widetilde{\mathcal{A}}_t^{\text{legal}} = \varnothing$ **then**

        **return** WAIT

    **end if**

    $\mathcal{A}_t^{\text{cand}} \leftarrow \mathcal{A}_t^{\text{pred}} \cap \widetilde{\mathcal{A}}_t^{\text{legal}}$

    **if** $\mathcal{A}_t^{\text{cand}} = \varnothing$ **then**

        **return** WAIT

    **end if**

    **for all** $a \in \mathcal{A}_t^{\text{cand}}$ **do**

        Compute $H_{\text{mix}}(\mathcal{H}_t, \mathcal{R}_t \cup \{a\})$

    **end for**

    **return** $a_{t+1}^\star \leftarrow \arg\min_{a \in \mathcal{A}_t^{\text{cand}}} \left( H_{\text{mix}}(\mathcal{H}_t, \mathcal{R}_t \cup \{a\}), \text{Pos}(\mathcal{A}_t^{\text{pred}}, a) \right)$

---

**Saved Step (SS).** We measure collaboration efficiency in two settings. In full-trajectory simulation, for each video $i$ we define $S_i = B_i - H_i$, where $B_i$ is the number of steps in the annotated human trajectory and $H_i$ is the number of steps actually executed by the human in simulation. We report

$$\text{SS} = \frac{1}{N} \sum_{i=1}^{N} S_i. \tag{10}$$

In the online one-step decision setting, each sample corresponds to a single decision point. We set $S_j = 1$ if the robot executes an action that belongs to the ground-truth trajectory (excluding `Terminate`), and $S_j = 0$ otherwise, and report $\text{SS} = \frac{1}{N_s} \sum_{j=1}^{N_s} S_j$.

**Parallel Action (PA).** We quantify the fraction of effective robot actions that advance a thread different from the human's most recent thread. Let $N^{\text{R}}$ be the total number of effective robot actions across all videos (excluding WAIT). For each

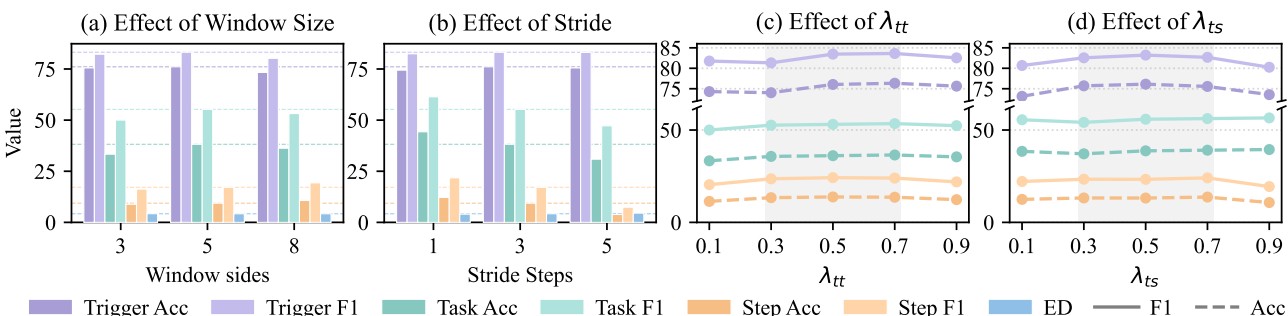

*Figure 6.* **Hyperparameter analysis on the mini set.** We evaluate the effect of window size, stride, and the loss weights $\lambda_{\mathrm{tt}}$ and $\lambda_{\mathrm{ts}}$. We report trigger/task/step accuracy and F1, as well as the future-action edit distance (ED; lower is better). Unless otherwise specified, we adopt window size 5, stride 3, and $\lambda_{\mathrm{tt}} = 0.3, \lambda_{\mathrm{ts}} = 0.5$.

*Table 4.* **Ablation of MLP projection heads for HBM.** Directly applying the binding loss to span-pooled hidden states yields lower validation loss and stronger overall performance.

| Method | Trig. mF1 | Task mF1 | Step mF1 | ED $\downarrow$ | Val Loss $\downarrow$ |
|---|---|---|---|---|---|
| w/o MLP (Ours) | **70.18** | **23.10** | 7.18 | **3.94** | **0.306** |
| w/ MLP ($d = 128$) | 67.55 | 21.41 | 6.99 | 4.05 | 0.649 |
| w/ MLP ($d = 256$) | 68.60 | 21.18 | **7.25** | 4.03 | 0.694 |

effective robot action $a$, let $h_{\mathrm{prev}}$ be the most recent human action before the decision. If $\pi(a) \neq \pi(h_{\mathrm{prev}})$, we count it as a parallel-action event. Let $P$ be the total number of such events, and define $\mathrm{PA} = P/N^{\mathrm{R}}$.

## B. Additional Quantitative Analyses

We provide additional quantitative analyses to better characterize the practical behavior of ProAct-Helper, including training-design ablations, trigger error modes, representation analyses, OOD generalization, and inference latency.

### B.1. Hyperparameter Analysis

We analyze the effects of temporal configuration and loss-weighting strategies in Figure 6. The results highlight a consistent trade-off between temporal coverage, noise accumulation, and cross-level regularization strength.

A window size of 5 achieves the best balance across trigger, task, and step prediction by providing sufficient temporal context to capture task state transitions without introducing excessive irrelevant frames. Smaller windows lack coverage for state evolution, while larger windows dilute critical cues and incur higher computational cost. Similarly, a stride of 3 yields the strongest performance by effectively reducing redundancy while preserving key temporal signals. Smaller strides introduce short-term noise, whereas larger strides miss informative transitions.

We further examine the task–trigger and task–step consistency losses. Moderate weighting consistently improves task and step recognition, indicating the benefit of cross-level regularization. However, overly large weights lead to performance degradation, suggesting that excessive emphasis on consistency suppresses fine-grained discriminative learning. Based on these observations, we adopt $\lambda_{\mathrm{tt}} = 0.3$ and $\lambda_{\mathrm{ts}} = 0.5$, which provide stable gains while maintaining balanced performance across all hierarchies.

### B.2. Projection Head Ablation

HBM applies the hierarchical binding loss directly to the span-pooled hidden states $H_\ell$. We additionally evaluate whether introducing a separate projection head, a common design in contrastive representation learning, improves cross-level binding. Specifically, we train two variants with a two-layer MLP projection head, $\mathrm{Linear}(H, d) \rightarrow \mathrm{ReLU} \rightarrow \mathrm{Linear}(d, d)$, where $H = 2048$ is the hidden size of Qwen2.5-VL and $d \in \{128, 256\}$. We use the same $1/8$ stratified mini-split as in Section 6.3 and keep all other training settings unchanged.

*Table 5.* **Under-triggering and over-triggering analysis on representative models.** FPR corresponds to over-triggering, while FNR corresponds to under-triggering.

| Method | FPR ↓ | FNR ↓ | mF1 ↑ |
|---|---|---|---|
| GPT-4o | 39.2 | 59.2 | 46.7 |
| Gemini-2.5-Pro | 77.4 | **9.2** | 55.9 |
| ProAct-Helper (7B) | **34.5** | 13.2 | **76.6** |

*Table 6.* **Cross-level representation consistency on the full test set.** HBM increases both CKA and cosine similarity between adjacent hierarchy levels.

| Model | $CKA(t,T)$ ↑ | $CKA(T,s)$ ↑ | $Cos(t,T)$ ↑ | $Cos(T,s)$ ↑ |
|---|---|---|---|---|
| w/o HBM | 0.628 | 0.821 | 0.197 | 0.406 |
| w/ HBM | **0.994** | **0.991** | **0.987** | **0.978** |

As shown in Table 4, adding a projection head does not improve HBM. Both MLP variants yield lower trigger and task mF1, comparable or lower step mF1, worse future-action ED, and substantially higher validation loss. This suggests that, in our autoregressive generation setting, directly binding the span-pooled hidden states provides a cleaner optimization path for cross-level alignment, whereas an additional randomly initialized projection head makes the joint optimization harder.

### B.3. Under-triggering and Over-triggering Analysis

Trigger detection is evaluated frame-wise over keyframe windows. To better characterize its error modes, we report false positive rate (FPR) and false negative rate (FNR) for representative models, where FPR measures over-triggering and FNR measures under-triggering. As shown in Table 5, GPT-4o exhibits a high FNR, indicating that it frequently misses intervention-worthy states, while Gemini-2.5-Pro exhibits a high FPR, indicating a tendency to produce unnecessary interventions. In contrast, ProAct-Helper achieves a better balance between the two error modes and obtains the highest macro-F1 among the compared representative models. This result suggests that trigger detection requires contextual intervention judgment rather than merely recognizing the ongoing action.

### B.4. Representation Analysis of HBM

We further analyze the representation effect of HBM to understand how cross-level binding improves the trigger–task–step hierarchy. ProAct-75 contains a highly imbalanced hierarchy, with only two trigger classes but many task and step classes. Under such long-tailed supervision, autoregressive token prediction alone may not sufficiently organize representations across hierarchy levels. HBM is designed to address this issue by binding parent–child representations while preserving fine-grained discrimination. Here $t$, $T$, and $s$ denote trigger, task, and step representations, respectively.

We first evaluate cross-level consistency using centered kernel alignment (CKA) and cosine similarity on the full test set. As shown in Table 6, HBM substantially increases both trigger–task and task–step consistency. Compared with the model without HBM, CKA increases from 0.628 to 0.994 for trigger–task and from 0.821 to 0.991 for task–step. Cosine similarity shows a similar trend, increasing from 0.197 to 0.987 for trigger–task and from 0.406 to 0.978 for task–step. These results indicate that HBM strengthens cross-level semantic alignment between adjacent hierarchy levels.

High cross-level similarity alone does not necessarily imply correct parent–child alignment, since it could also arise from indiscriminate global similarity. To examine this, we randomly shuffle the parent–child pairing at test time and recompute cosine similarity between mismatched hierarchy levels. As shown in Table 7, without HBM, shuffling only changes trigger–task cosine similarity marginally, with a paired-shuffled gap of 0.009. With HBM, the paired-shuffled gap becomes much larger, reaching 0.768 for trigger–task and 0.780 for task–step. This suggests that the increased similarity induced by HBM is tied to the correct hierarchical correspondence rather than indiscriminate global similarity.

We further examine whether the stronger cross-level structure is obtained through trivial representation collapse. We compute effective rank (eRank), which measures how broadly information is distributed across the hidden dimensions, and shared subspace overlap (SSO50), which measures the fraction of task-level variance explained by the top-50 principal components of trigger representations. As shown in Table 7, HBM increases SSO50 from 7.7% to 64.7%, indicating a much stronger shared cross-level subspace. Meanwhile, eRank also increases from 637 to 934, suggesting that the representation remains high-dimensional rather than collapsing into a narrow subspace.

*Table 7.* **Pair-specific alignment and representation geometry under HBM.** Pair and Shuf denote paired and shuffled cosine similarities on the full test set. SSO50 measures the fraction of task-level variance explained by the top-50 principal components of trigger representations.

| | Paired and shuffled cosine similarity | | | | | | Geometry | |
| Model | $\text{Pair}_{t,T}$ | $\text{Shuf}_{t,T}$ | $\Delta$ | $\text{Pair}_{T,s}$ | $\text{Shuf}_{T,s}$ | $\Delta$ | eRank $\uparrow$ | SSO50 $\uparrow$ |
|---|---|---|---|---|---|---|---|---|
| w/o HBM | 0.197 | 0.188 | 0.009 | 0.406 | 0.219 | 0.187 | 637 | 7.7% |
| w/ HBM | **0.987** | 0.220 | **0.768** | **0.978** | 0.199 | **0.780** | **934** | **64.7%** |

*Table 8.* **View-set OOD evaluation on Ego-Exo4D and Our selected videos.** Models are trained on the **Best View** training set and evaluated on **Best View** test set or **Other View** test set (all non-best views). We report macro-averaged (mAcc/mF1) and micro-averaged (Acc/F1) metrics for trigger/task/step classification and future action edit distance (ED; lower is better).

| Dataset | View Set | Trigger (%) | | | | Task (%) | | | | Step (%) | | | | ED $\downarrow$ |
|---|---|---|---|---|---|---|---|---|---|---|---|---|---|---|
| | | mAcc | mF1 | Acc | F1 | mAcc | mF1 | Acc | F1 | mAcc | mF1 | Acc | F1 | |
| | | *ProAct-Helper (based on Qwen2.5-VL-3B-Instruct)* | | | | | | | | | | | | |
| Ego-Exo4D | Other View | 54.40 | 63.48 | 89.61 | 94.37 | 0.90 | 1.62 | 5.76 | 10.89 | 0.77 | 1.36 | 6.49 | 12.18 | 4.41 |
| | Best View | 61.22 | 71.32 | 92.09 | 95.72 | 35.13 | 43.67 | 60.68 | 75.53 | 19.56 | 24.96 | 20.68 | 34.27 | 3.53 |
| Ours | Other View | 62.85 | 76.17 | 81.45 | 87.38 | 17.83 | 23.43 | 35.02 | 51.87 | 10.02 | 13.92 | 23.86 | 38.53 | 2.96 |
| | Best View | 85.51 | 92.11 | 93.23 | 95.09 | 81.78 | 89.21 | 83.26 | 90.86 | 62.40 | 69.76 | 73.57 | 84.77 | 0.78 |
| | | *ProAct-Helper (based on Qwen2.5-VL-7B-Instruct)* | | | | | | | | | | | | |
| Ego-Exo4D | Other View | 54.15 | 63.40 | 88.97 | 93.99 | 0.62 | 1.10 | 6.10 | 11.51 | 0.95 | 1.67 | 7.17 | 13.38 | 4.33 |
| | Best View | 60.68 | 70.64 | 92.18 | 95.79 | 21.38 | 26.27 | 62.73 | 77.10 | 19.96 | 25.95 | 22.30 | 36.47 | 3.50 |
| Ours | Other View | 63.74 | 76.82 | 82.35 | 88.15 | 31.41 | 39.53 | 50.74 | 67.32 | 16.75 | 21.30 | 29.20 | 45.20 | 2.55 |
| | Best View | 83.77 | 91.05 | 92.44 | 94.58 | 84.44 | 91.12 | 84.48 | 91.59 | 63.64 | 71.04 | 70.65 | 82.80 | 0.82 |

These results show that HBM strengthens correct parent–child alignment while avoiding trivial representation collapse. This supports the design motivation of applying explicit cross-level binding to the trigger–task–step hierarchy under long-tailed supervision.

## B.5. Cross-View Generalization

In this view-set OOD evaluation, we train all models on the Best View training set and evaluate them on both the Best View test set and the Other View test set, which contains all non-best views, to characterize generalization robustness under systematic viewpoint shifts. As shown in Table 8, despite the more challenging distribution shift in Other View due to variations in perspective, occlusion, and visibility that destabilize visual cues, ProAct-Helper maintains substantially more reliable task and step recognition as well as future action prediction. For task and step detection on Qwen2.5-VL-7B, our method's Task F1 decreases from 91.59 on Best View to 67.32 on Other View, retaining approximately 73% of performance, whereas Ego-Exo4D drops from 77.10 to 11.51, retaining only 15%. For Step F1, our method declines from 82.80 to 45.20, preserving roughly 55%, while Ego-Exo4D falls from 36.47 to 13.38, retaining merely 37%. Similar trends appear with the 3B backbone: our Task F1 on Other View remains at 51.87, significantly outperforming Ego-Exo4D's 10.89.

From a long-tail perspective measured by macro-averaged metrics, our method achieves Task mF1 of 23.43 and 39.53 on the 3B and 7B backbones respectively on Other View, compared to 1.62 and 1.10 for Ego-Exo4D. Step mF1 similarly reaches 13.92 and 21.30 for our method against 1.36 and 1.67 for Ego-Exo4D, demonstrating superior robustness on tail categories under viewpoint shifts. Finally, for future action prediction, our method exhibits lower edit distance on Other View: 2.96 compared to 4.41 on 3B, and 2.55 compared to 4.33 on 7B, further validating more consistent procedural prediction under suboptimal viewing conditions. Overall, these results demonstrate that our approach not only achieves higher performance ceilings on Best View but also maintains superior cross-view stability and transferability in more realistic deployment scenarios with non-optimal viewpoints.

*Table 9.* **Actor-split OOD evaluation on self-collected videos.** The OOD setting trains on Actors 1–4 and evaluates on the held-out Actor 5, while the full-data setting trains with videos from all five actors.

| Setting | Trig. mAcc | Trig. mF1 | Trig. Acc | Trig. F1 | Task mAcc | Task F1 | Step mAcc | Step F1 | Future ED ↓ |
|---|---|---|---|---|---|---|---|---|---|
| Actor-split OOD | 60.64 | 74.33 | 80.06 | 86.46 | 29.51 | 64.21 | 11.04 | 34.97 | 2.670 |
| Full-data training | 75.51 | 85.57 | 89.39 | 93.00 | 88.15 | 94.64 | 79.24 | 89.97 | 0.563 |

*Table 10.* **Average wall clock latency per timestep.** Perception aggregates trigger, task, step, and future prediction, while Planning denotes online one-step action selection.

| Base model | Perception time (s) | Planning time (s) |
|---|---|---|
| *Open-source MLLMs* | | |
| Qwen3-VL-30B-A3B-Instruct | 15.04 | 0.51 |
| Qwen3-Omni-30B | **1.20** | 0.52 |
| Qwen2.5-VL-32B-Instruct | 46.04 | 0.29 |
| Qwen2.5-VL-3B-Instruct | 12.48 | 0.16 |
| *ProAct-Helper (based on Qwen2.5-VL-7B-Instruct)* | | |
| ProAct-Helper | 2.75 | **0.08** |

### B.6. Actor-Split OOD Generalization

We further evaluate out-of-distribution generalization across actors on our self-collected videos. Specifically, we train ProAct-Helper with the Qwen2.5-VL-3B backbone on videos from Actors 1–4 and evaluate it on the held-out Actor 5. As a reference, we also report a full-data setting trained with videos from all five actors under the same training protocol. This evaluation isolates actor-level distribution shift, where the task set and annotation protocol remain fixed but execution style, temporal rhythm, body motion, and interaction details vary across individuals.

As shown in Table 9, the actor-split setting leads to a moderate drop in trigger detection, with Trigger F1 decreasing from 93.00 to 86.46. The degradation is more pronounced for fine-grained perception: Task F1 decreases from 94.64 to 64.21, and Step F1 decreases from 89.97 to 34.97. Future-action prediction is also affected, with ED increasing from 0.563 to 2.670. These results suggest that trigger-level intervention cues transfer more reliably across actors than fine-grained task and step states. Cross-actor generalization therefore remains a challenging setting in ProAct-75, especially for detailed procedural understanding and future-step prediction.

### B.7. Inference Latency Comparison

We report wall clock inference latency for the proactive pipeline, comparing ProAct-Helper-7B with representative open-source MLLMs. Following the evaluation protocol, Perception aggregates the first four perception subtasks, including trigger detection, task identification, step detection, and short-horizon future action prediction. Planning denotes online one step decision making for proactive action selection under task-graph constraints, where the policy outputs a single executable action at each timestep.

Table 10 shows that ProAct-Helper-7B substantially reduces end-to-end latency compared to general-purpose open-source baselines, especially on the perception stack. The planning latency of ProAct-Helper-7B remains low, which is critical for interactive deployment where decisions must be produced continuously over keyframe windows.

## C. Qualitative Results

We present qualitative examples to illustrate typical success and failure patterns of the proposed proactive response pipeline. These cases complement the quantitative results by revealing how cross-stage consistency and task-graph constraints manifest in real videos. We additionally visualize the results of proactive action selection tasks, illustrating how graph-feasible robot actions are chosen from short-horizon candidates.

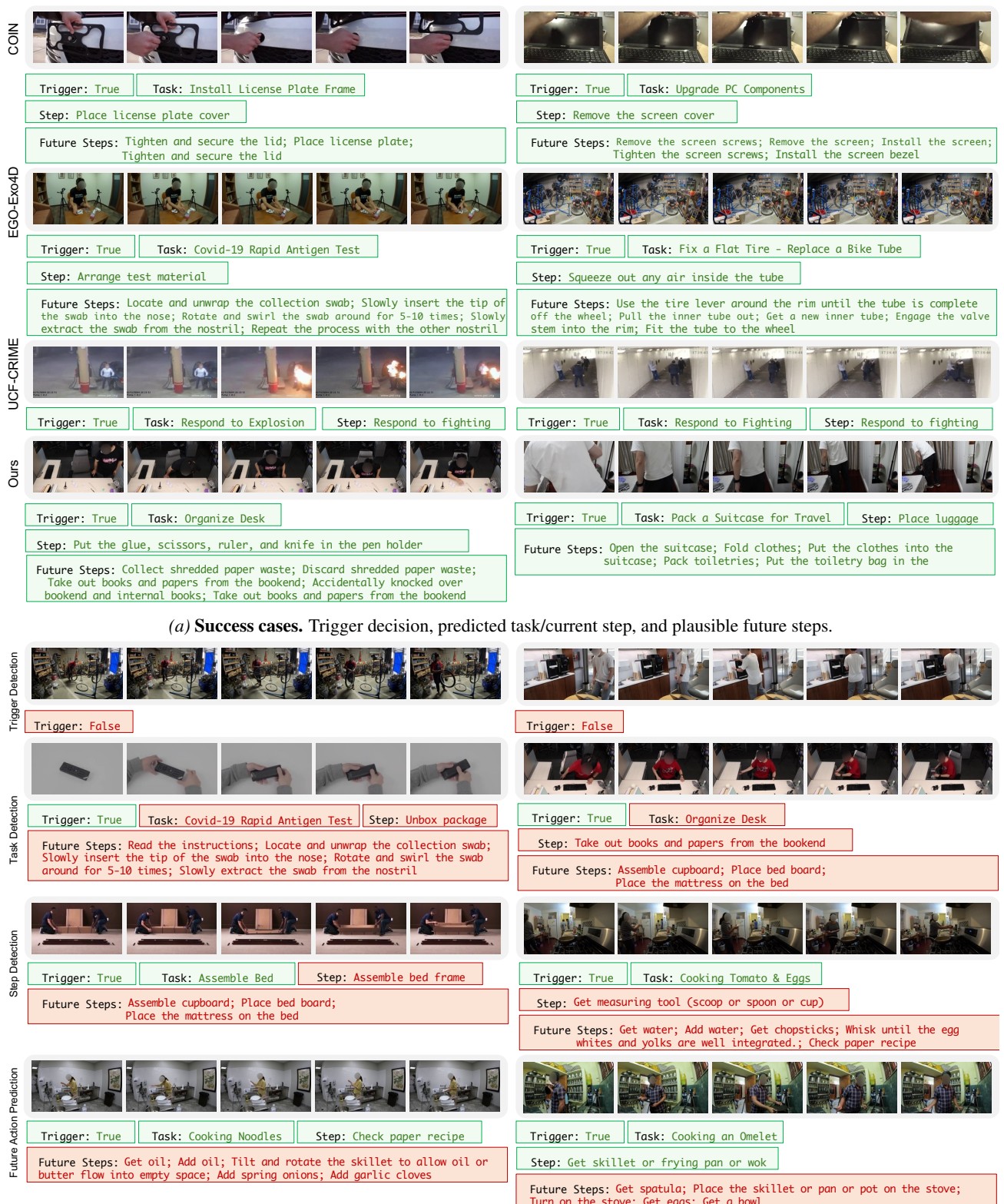

*(a)* **Success cases.** Trigger decision, predicted task/current step, and plausible future steps.

*(b)* **Failure cases.** Task/step confusion, invalid or repetitive futures, and incomplete generations.

*Figure 7.* **Qualitative results across sources.** We show representative success and failure patterns for proactive prediction and future-step generation.

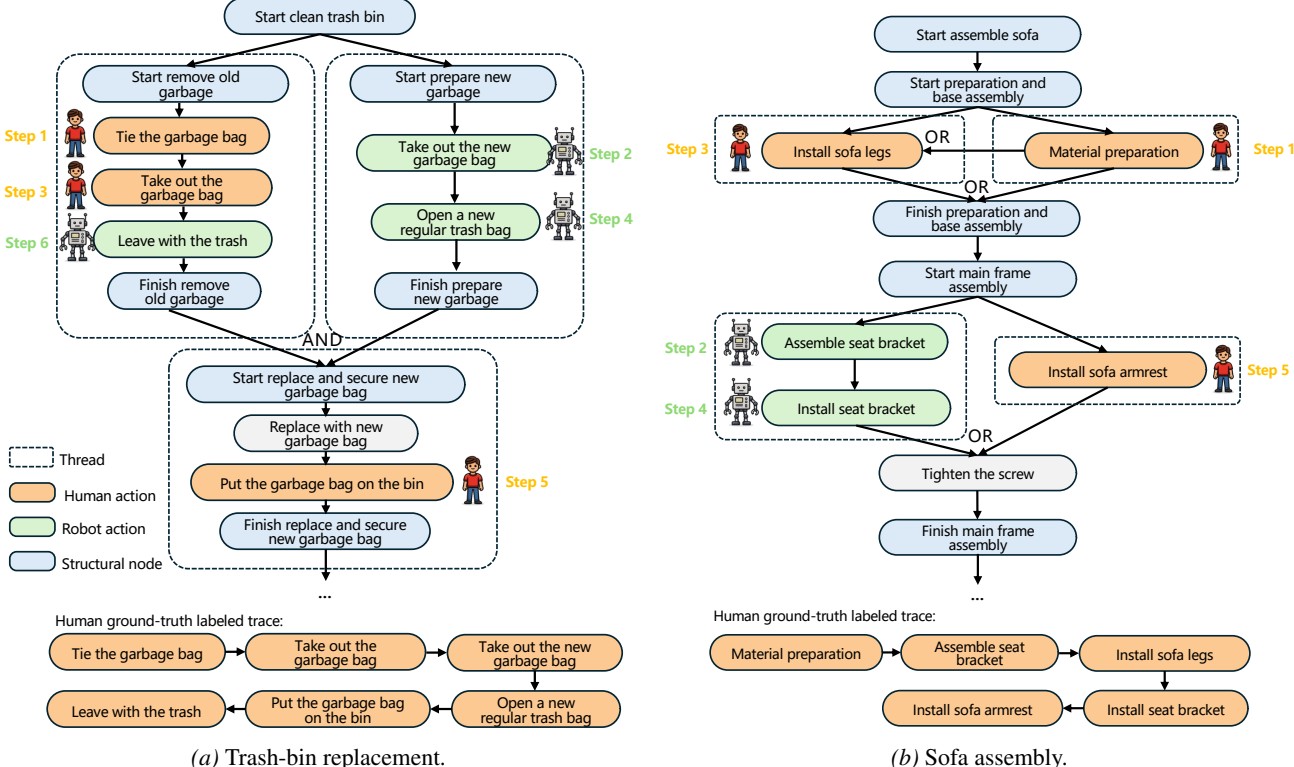

*(a)* Trash-bin replacement.    *(b)* Sofa assembly.

*Figure 8.* **Stage-5 visualization of proactive action selection under task-graph constraints.** We overlay the human ground-truth trace on the annotated task graph and visualize how the agent selects the next proactive action after short-horizon future-step generation. Candidate actions are first filtered by graph feasibility (*e.g.*, prerequisite satisfaction and AND/OR dependencies) and then ranked to choose an actionable, procedure-aligned robot step that best supports the ongoing workflow. Left: trash-bin replacement; right: sofa assembly.

## C.1. Success Cases

**State Detection and Prediction Tasks.** Figure 7a shows representative success cases of our proactive pipeline across multiple sources (COIN, Ego-Exo4D, UCF-Crime, and our collected data). Overall, the model demonstrates strong cross-stage consistency: it triggers interventions when warranted, identifies the correct task context and ongoing step, and generates plausible short-horizon future steps that remain coherent with the observed workflow. Notably, the predicted futures are typically actionable and procedurally aligned (*e.g.*, preparing tools before execution), suggesting that the model has learned transferable procedural priors beyond dataset-specific appearance cues.

**Proactive Action Selection.** Beyond per-stage predictions, Figure 8 visualizes how the pipeline instantiates graph-constrained decision making at the final stage. Given the recognized task/step context and the short-horizon candidates, the agent filters out graph-infeasible actions (*e.g.*, violated prerequisites) and selects an actionable next robot step that is procedurally aligned with the ongoing workflow. As shown in the trash-bin replacement and sofa assembly examples, the selected actions tend to prioritize prerequisite- and preparation-type steps before execution, demonstrating effective use of task-graph constraints for low-risk and high-utility interventions.

## C.2. Failure Cases

Figure 7b summarizes typical failure modes. First, the model may *confuse semantically related tasks* under limited visual evidence (*e.g.*, mapping emergency response contexts to an incorrect but plausible category), which then cascades to step-level mismatch. Second, we observe *procedural hallucination and redundancy* in future-step generation, such as repeated steps or inserting irrelevant sub-procedures, indicating that language priors can dominate when the visual state is ambiguous. Third, some outputs become *graph-inconsistent or incomplete* (*e.g.*, truncated step descriptions), which suggests remaining challenges in maintaining well-formed, graph-feasible plans under long-tail or OOD conditions. These cases motivate incorporating stricter graph-constrained decoding and validity-aware training objectives to further suppress

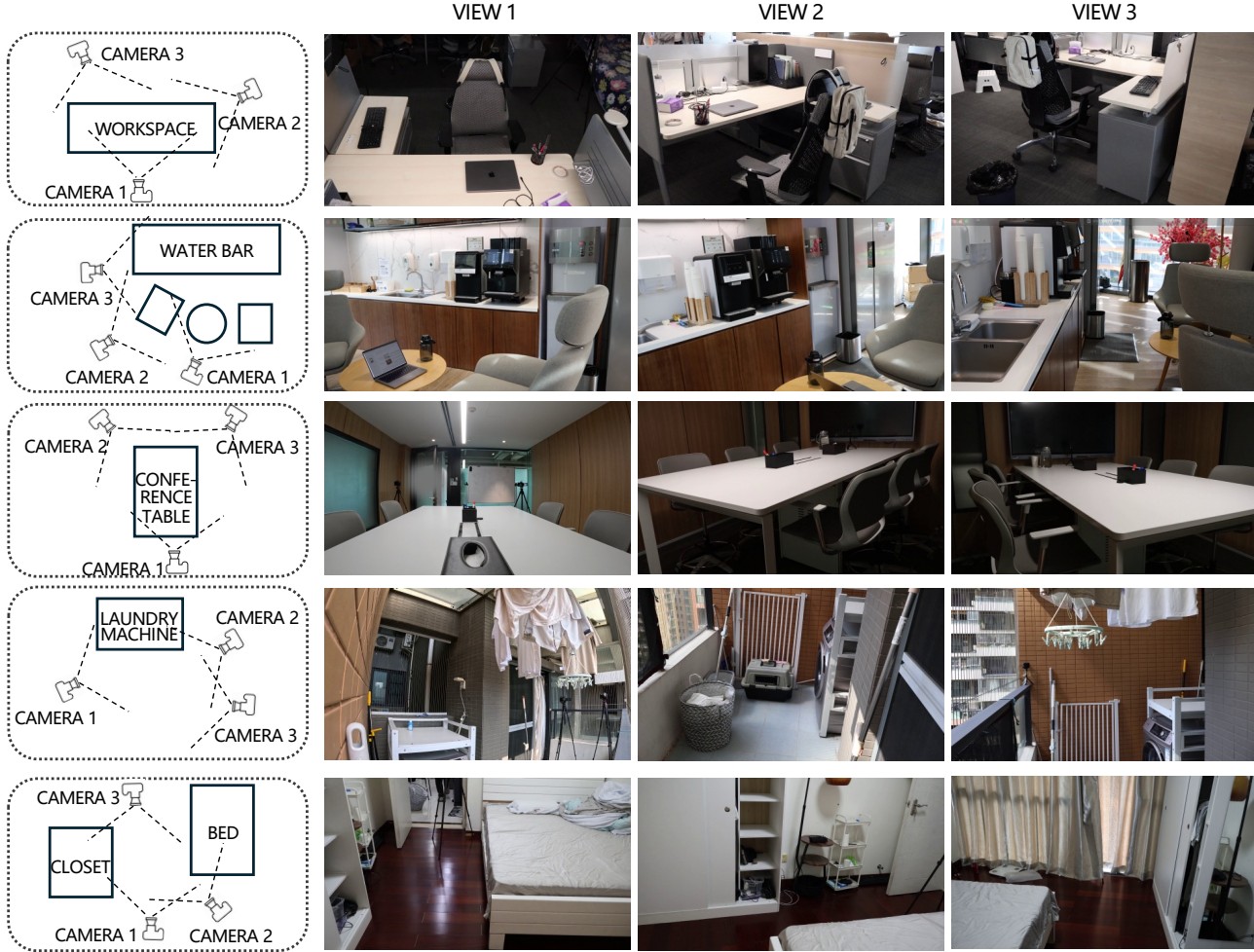

*Figure 9.* **Multi-view scene setups and example frames.** The left column illustrates the camera layouts for each scene, while the right columns show representative synchronized frames from the three viewpoints (View 1–3).

invalid or low-utility interventions. In addition, errors in upstream task/step recognition can lead to *mis-constrained action selection*, where graph filtering over-prunes feasible actions or favors an incorrect thread, yielding conservative or suboptimal interventions.

## D. Dataset and Annotation Details

We provide additional details of the multi-view data collection setup and quality control. Each scene is captured by three synchronized cameras (Canon M50) at 4K/30fps to provide complementary viewpoints and reduce occlusion. For our self-collected multi-view videos, we additionally select a Best View based on manipulation visibility and minimal occlusion to form a standardized evaluation set. All videos are annotated under a unified step-level protocol, where boundaries correspond to semantic changes in human actions. Quality control is conducted in multiple rounds, and disagreements are resolved through expert discussion to ensure consistent step boundaries and labels across sources.

# E. Task-Graph Gallery

To facilitate reproducibility and error analysis, we include a gallery visualization of all annotated task graphs. In these visualizations, tasks are color-coded by scenario: Assistance ▪, Maintenance ▪, and Safety Monitoring ▪, with overlapping regions indicating tasks that belong to both Assistance and Maintenance scenarios. Besides the sunburst renderings, we also report the distribution over mid-level nodes for each task graph, which characterizes how execution threads are induced by mid-level groups. We note that the optional priority scores are not used for training, since each task has fixed scores in our closed-set setting and can be directly retrieved via a lookup table when needed.

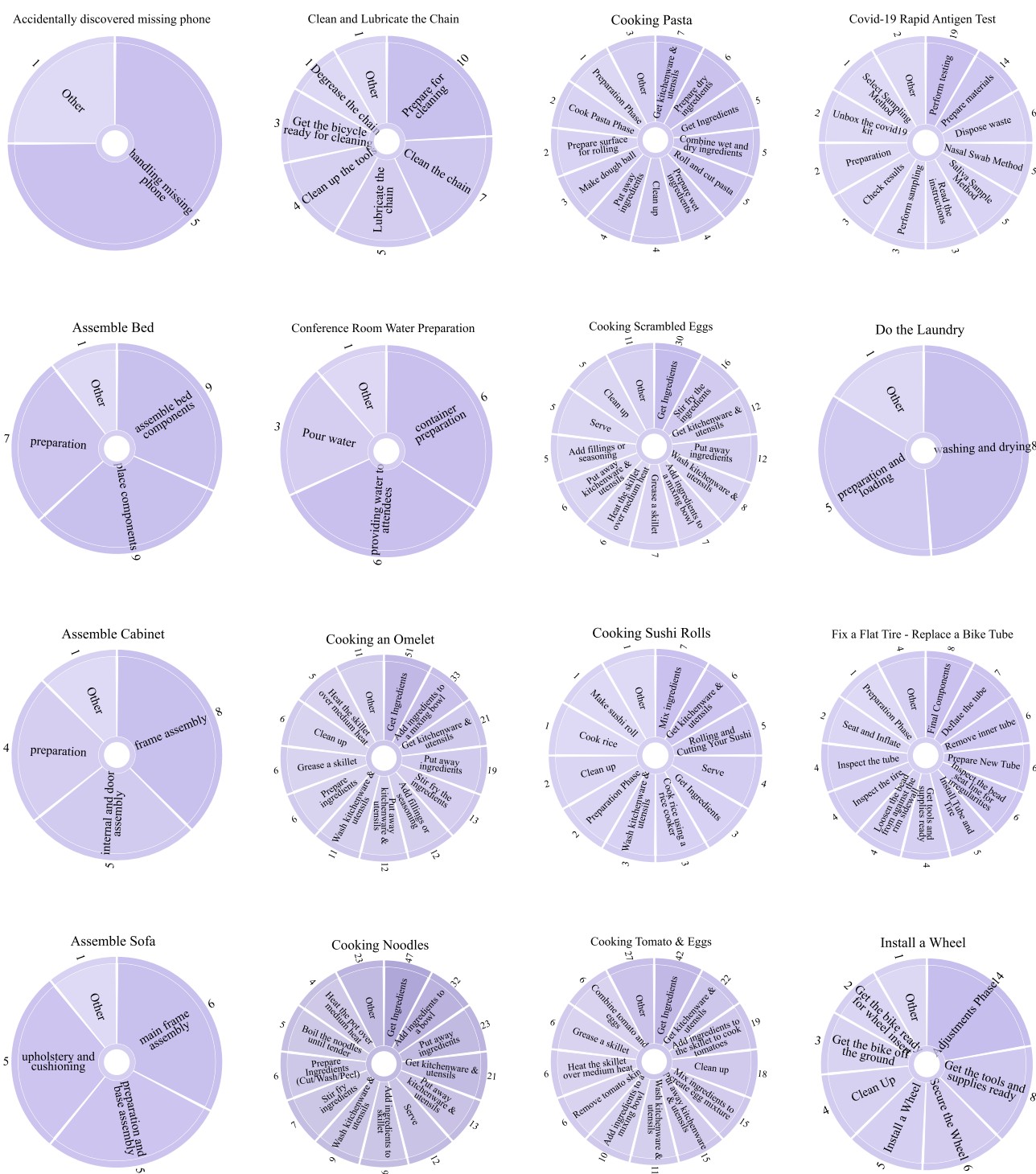

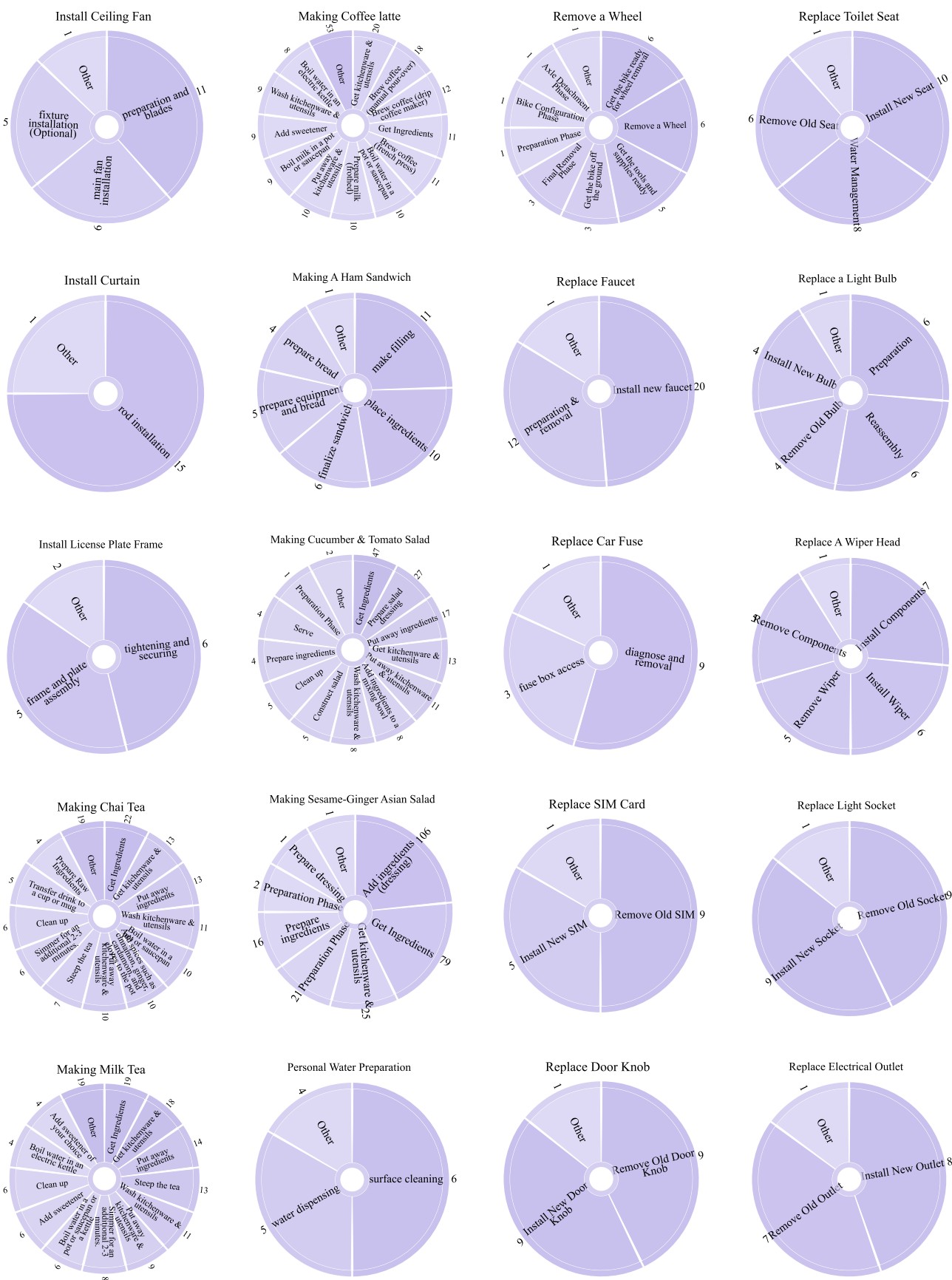

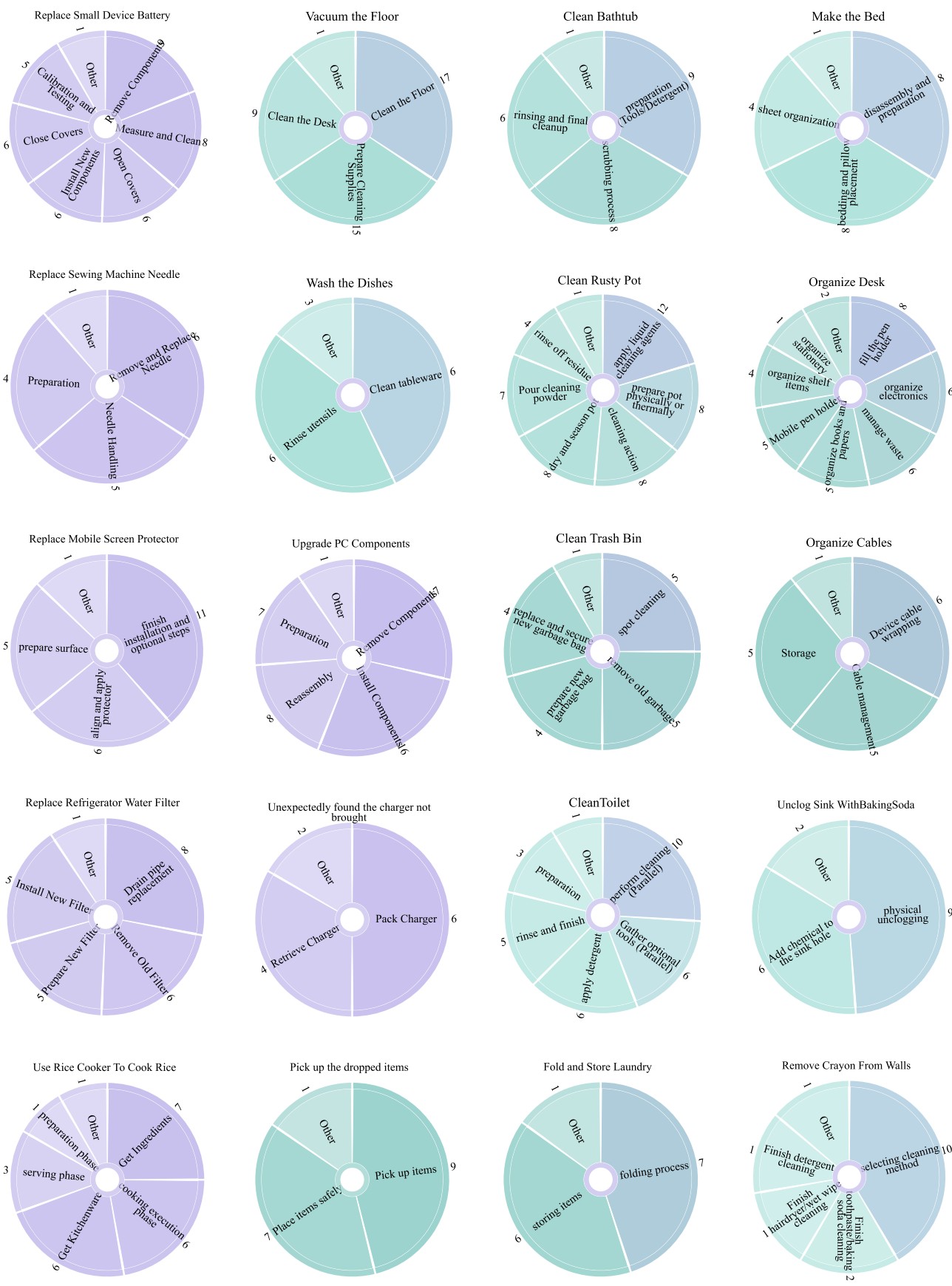

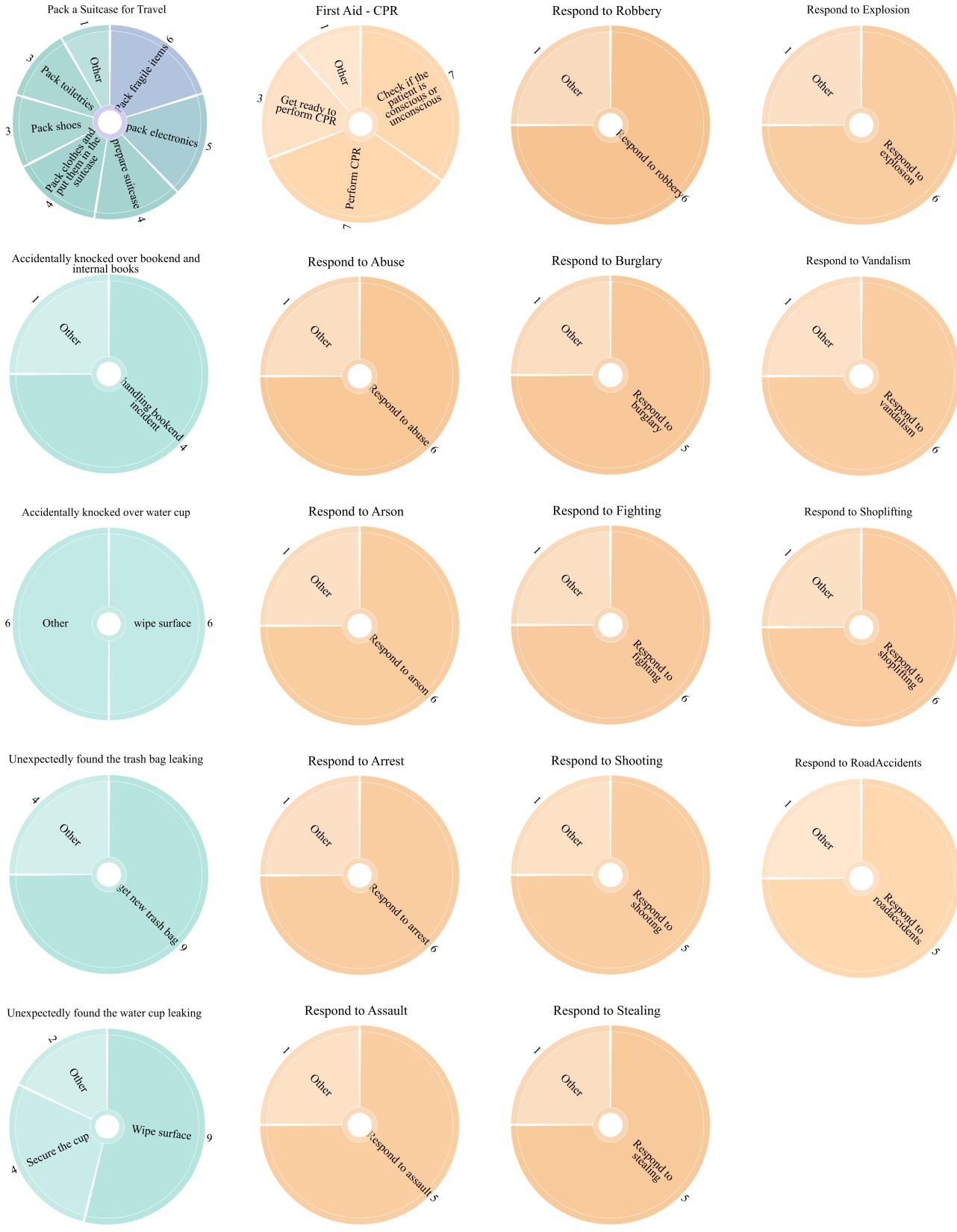

*Figure 10.* **Task-graph collection (75 tasks).** We visualize the annotated task graphs for all tasks as sunburst-style DAG renderings. Each sector denotes a step, radial depth indicates hierarchical decomposition, and connectivity implicitly reflects prerequisite and branching constraints.

## F. Prompt Templates

In this section, we document the prompt templates used in our experiments. We include (i) the *proactive action selection* prompt, which takes the task graph, progression state, human next-step horizon, and a candidate action set as input and requires a strict JSON action output, and (ii) the two-stage *state perception* prompts for trigger/task and step/future prediction from video frames. We also provide representative example inputs and raw outputs from several LLMs to illustrate common failure modes (*e.g.*, malformed JSON, truncated generations, and constraint violations) under identical prompting.

---

**Proactive Action Selection Prompt (Template)**

```
System:
You are a robot action planner collaborating with a human on a procedural task.
Choose exactly ONE next robot action from the provided candidate list.

Hard constraints:
1) You must output a JSON object with key "action".
2) The value must be exactly one string from CANDIDATE_ACTIONS, or "Wait / None" if
   the list is empty.
3) Prefer actions that are parallel to the human's current thread (i.e., from a
   different execution thread in the task graph), when possible. Thread (thread): an
   independent branch in the task graph induced by the same mid-level start/end node
   pair; different threads have no shared nodes.

User:
TASK: {task_name}

TASK_GRAPH (compact):
{task_graph}

COMPLETED_STEPS:
{completed}

HUMAN_IMMEDIATE_NEXT (do NOT do these):
{human_immediate}

HUMAN_FUTURE_HORIZON:
{human_future}

CANDIDATE_ACTIONS:
{candidates}

Return JSON only:
{"action":"...","reason":"...","confidence":0-1}
```

---

**Planning Example Input (State ID: COIN\_-2CxSAVwFqE:1)**

```
Task: Upgrade PC Components
Candidate actions:
[
  "Remove the old hard drive",
  "Install the hard drive",
  "Install laptop back cover",
  "Terminate"
]
Note: Logged "action" is the final executed robot action.
```

```
If a proposed action's preconditions are not met, it becomes "Wait / None"
and the reason is appended with "| prereq_not_met".
```

## Planning Output (GPT-4o)

```
{
  "action": "Wait / None",
  "reason": "This action follows directly after 'Remove the old hard drive', which is
    in the HUMAN_IMMEDIATE_NEXT, allowing continuous progress without interruption. |
    prereq_not_met",
  "confidence": 0.9
}
(raw proposed action: "Install the hard drive")
```

## Planning Output (Gemini-2.5-Flash)

```
{
  "action": "Wait / None",
  "reason": "The human is about to",
  "confidence": null
}
```

## Planning Output (Qwen3-Max)

```
{
  "action": "Wait / None",
  "reason": "The human is currently focused on removing the old hard drive (
    HUMAN_IMMEDIATE_NEXT), so I should avoid that step. Installing the new hard drive
    is a valid candidate action not in HUMAN_IMMEDIATE_NEXT, and it unlocks the next
    human steps while reducing thread switching. | prereq_not_met",
  "confidence": 0.95
}
(raw proposed action: "Install the hard drive")
```

## Planning Output (DeepSeek-v3.2)

```
{
  "action": "Wait / None",
  "reason": "All candidate actions are in HUMAN_IMMEDIATE_NEXT (Remove the old hard
    drive) or would violate constraint #3. According to hard constraint #5, I must
    choose 'Wait / None' when every candidate action is in HUMAN_IMMEDIATE_NEXT or the
    list is effectively empty for the robot to act upon.",
  "confidence": 1.0
}
```

**Proactive response Stage-1 Prompt (Trigger + Task)**

```
System:
You are a precise vision-language model for video task detection.
From the given frames (oldest->latest), decide CURRENT high-level task.
Return strict JSON only: {"is_trigger": <true_or_false>, "task": "<
    one_of_list_or_others>"}.

IMPORTANT:
- If the person needs help with ANY task from the list, set is_trigger=true and output
    that task; otherwise set is_trigger=false.
- Avoid false negatives: do NOT output {"is_trigger": false, "task": "others"} unless
    none matches.

User:
Frames: <frame_descs joined by comma>
Candidate tasks: <comma-joined task list>
If no task fits, use 'others' and set is_trigger=false.
```

**Proactive response Stage-2 Prompt (Step + Future Steps + Scores)**

```
System:
You are a precise vision-language model for fine-grained procedural understanding.
From the frames and the current task, decide CURRENT step and the NEXT 5 likely future
    steps.
All steps MUST be chosen from the provided list. Scores must be integers 1..3 and
priority = max(urgency, value).
Return strict JSON only:
{
  "current_step": "<one_step_from_list_or_empty>",
  "future_steps": ["<step1>","<step2>","<step3>","<step4>","<step5>"],
  "scores": { "urgency": <int>, "value": <int>, "priority": <int> }
}
```

**Proactive response Example Input (EGO\_EXO4D\_...\_cam02)**

```
Frame window:
[
  "[idx=0 t=0.00s]",
  "[idx=32 t=1.28s]",
  "[idx=51 t=2.04s]",
  "[idx=67 t=2.68s]"
]
GT: task = "Covid-19 Rapid Antigen Test"
    step = "Read the instructions"
    is_trigger = true
```

**Proactive response Output (GPT-4o)**

```
{
  "pred_is_trigger": false,
  "pred_task": "others",
  "pred_step": "",
```

```
    "pred_future_steps": [],
    "scores": {}
}
```

## Proactive response Output (Gemini-2.5-Flash)

```
{
  "pred_is_trigger": false,
  "pred_task": "others",
  "pred_step": "",
  "pred_future_steps": [],
  "scores": {}
}
```

## Proactive response Output (Qwen3-VL-Plus)

```
{
  "pred_is_trigger": true,
  "pred_task": "Organize Desk",
  "pred_step": "Take the cloth",
  "pred_future_steps": [
    "Wipe the desk with a tissue",
    "Place the item in a dry area",
    "Support book upright",
    "Discard tissue",
    "Place the book in a dry area"
  ],
  "scores": { "urgency": 1, "value": 1, "priority": 1 }
}
```

*Figure 11.* **Prompts and example outputs for ProAct-75.** Each card is rendered as a web-style panel (full-width title bar) and may span pages automatically.

