# OpenReview forum: "ProAct: A Benchmark and Multimodal Framework for Structure-Aware Proactive Response"
_ICML.cc/2026/Conference — ICML 2026 regular_

### Official Review · Reviewer_Hyas · 2026-03-03

**Soundness:** 3
**Presentation:** 2
**Significance:** 2
**Originality:** 2
**Overall Recommendation:** 3
**Confidence:** 2

**Summary:**

This paper introduces ProAct-75, a benchmark for proactive response, together with a baseline framework, ProAct-Helper. The dataset provides explicit DAG-based task graphs with AND/OR dependencies and parallel threads, and defines five tasks: trigger detection, task detection, step detection, future action prediction, and proactive action selection.

**Compliance With Llm Reviewing Policy:**

Affirmed.

**Key Questions For Authors:**

Is the entropy-based strategy consistently effective across different task structures? Are there counterexamples?

How robust is the method to noise or errors in the task graph?

Is there representation-level evidence that HBM improves cross-level consistency?

How does the heuristic planner compare with learning-based planning approaches?

**Limitations:**

The framework relies on high-quality task graphs, which may be difficult to obtain in practice.

The planning module is not learned, which may limit scalability.

There is no evaluation in real-world robotic settings.

Broader cross-domain generalization remains to be demonstrated.

**Strengths And Weaknesses:**

Strengths:

The problem is meaningful and well-motivated, emphasizing structure-aware proactive collaboration.

The benchmark is carefully constructed, and the explicit task graphs are a clear contribution.

Experiments are comprehensive, including ablations, simulation studies, and failure analysis.

The overall framework is coherent and interpretable.

Weaknesses:

The methodological novelty is moderate; both HBM and the entropy-based strategy are largely engineering-driven.

The action selection module is heuristic and not learned, with limited analysis of optimality.

There is no real-robot validation; evaluation is based on offline videos and simulation.

The fairness of comparisons (fine-tuned model vs. closed-source models) could be further clarified.

---

> ### Author Rebuttal · Authors · 2026-03-29
>
> ### W1 & Q3: Methodological novelty and HBM cross-level evidence
> HBM borrows the contrastive formulation but does not directly apply standard contrastive learning. The trigger-task-step hierarchy has severe class imbalance (2 trigger classes, 75 tasks, 1,251 steps), and autoregressive supervision alone cannot enforce cross-level consistency. HBM binds parent-child representations to improve rare-class separability. Table 2 confirms each binding independently contributes (+2.82% Task mAcc, +2.70% Step mAcc). We provide representation-level evidence using CKA [1] and cosine similarity on the full test set.
>
> **Table R6. Cross-level representation consistency.**
> | Model | CKA(t,T)↑ | CKA(T,s)↑ | Cos(t,T)↑ | Cos(T,s)↑ |
> |-------|:---------:|:---------:|:---------:|:---------:|
> | w/o HBM | 0.628 | 0.821 | 0.197 | 0.406 |
> | w/ HBM | **0.994** | **0.991** | **0.987** | **0.978** |
>
> With HBM, both CKA and cosine similarity increase substantially, showing stronger cross-level parent-child alignment.
> ### W2, Q4 & L2: From heuristic planning to learned planning
> Please see our response to Reviewer Xfmg-W3 for the full analysis. Briefly, we use heuristic-generated SFT labels as a pragmatic choice at this stage, and further provide a new end-to-end variant that predicts all five tasks, including action selection.
>
> **Table R7. End-to-end comparison (3B).**
> | Method | Trig. F1 | Task F1 | Step F1 | ED↓ | SS | PA |
> |--------|---------|---------|---------|-----|------|-------|
> | w/ heuristic | 85.23 | 67.58 | 31.49 | 3.96 | 0.366 | 17.44 |
> | End-to-end | 85.56 | 62.86 | 30.35 | 4.23 | 0.348 | 16.85 |
>
> The end-to-end model achieves comparable perception and planning without heuristics. **This suggests that the end-to-end model can learn to imitate the heuristic policy.** We also note that SS and PA depend on upstream perception quality, so both could further improve with more accurate task and step prediction.
>
> ### W3 & L3: Real-robot validation
> Real-robot validation is outside the current scope. Our benchmark already evaluates perception and action selection on real-world videos captured from diverse viewpoints, while the simulation (Section 6.4) provides controlled evaluation under realistic human behavior traces. Real-robot deployment can build on top of this system, and because all training data comes from real-world videos, the sim-to-real gap is partially reduced. This remains an important direction for future work.
> ### W4 & L1: Fairness of comparisons and task graph availability.
> Mixing fine-tuned and zero-shot evaluation is standard in benchmarks such as MVBench [2]. We follow this convention and report a plain variant without HBM to isolate its contribution. Our benchmark provides an evaluation suite with ground-truth task graphs built via a semi-automated pipeline (Section 4.2). In some practical deployments, task graphs already exist as standard operating procedure documentation.
> ### Q1 & Q2: Robustness to graph structure variation
> We address Q1 and Q2 together as both concern graph robustness. The entropy heuristic handles diverse structures by design, selecting the single legal action for sequential graphs and minimizing thread-mixing entropy for parallel ones.
>
> To test noise robustness, we perturb graphs with three operations at 5% and 10% node ratios: APT inserts new parallel branches, RPT deletes existing ones, and AS adds skip connections without creating new threads (Section 3.1). We report SS (Saved Steps) as it measures outcome efficiency independent of graph topology, unlike PA whose definition of "parallel" varies with the graph itself.
>
> **Table R8. Robustness to task graph noise.**
> | Model | Original | APT(5%) | RPT(5%) | AS(5%) | APT(10%) | RPT(10%) | AS(10%) |
> |---------|:--------:|:-------:|:-------:|:------:|:--------:|:--------:|:-------:|
> | ProAct-Helper (3B) | 0.366 | 0.366 | 0.367 | 0.366 | 0.367 | 0.367 | 0.366 |
> | ProAct-Helper (7B) | 0.361 | 0.361 | 0.361 | 0.361 | 0.361 | 0.361 | 0.361 |
>
> SS remains virtually unchanged because the heuristic ranks actions by relative thread distribution rather than exact topology, and the intersection filter ensures only legal actions under the modified graph are considered. On countercases, 7.87% of robot actions involve symmetric threads where entropy loses discriminative power but causes no disruption, and 1.5% involve preemption of the human's next step while still saving a step. These cases suggest that a multi-step lookahead planner could further improve coordination, which is a promising direction for learned planning.
>  ### L4: Broader generalization
> We provide broader OOD evidence in our response to Reviewer YwTj-W1. Across cross-view, actor-split, and dataset-split OOD, trigger F1 drops by only 3.23% in the strictest setting.
>
> ### References
> [1] Kornblith et al., Similarity of Neural Network Representations Revisited, 2019, ICML.
>
> [2] Li et al., MVBench: A Comprehensive Multi-modal Video Understanding Benchmark, 2024, CVPR.

---

> > ### Author Rebuttal · Reviewer_Hyas · 2026-04-03
> >
> > Table R6 only shows the ablation results and the effectiveness of HBM, but I would have liked to see why it was effective.
> >
> > All other questions have been resolved.

---

> > > ### Author Response · Authors · 2026-04-03
> > >
> > > We sincerely thank the reviewer for the careful follow-up and for giving us the opportunity to clarify this point further. The key question here is why HBM is particularly effective in our setting.
> > >
> > > **We believe this is mainly because** HBM explicitly strengthens correct parent-child alignment without sacrificing fine-grained discrimination. This is particularly important in the trigger-task-step hierarchy of ProAct-75, where severe long-tail imbalance and pure autoregressive supervision make cross-level structure difficult to learn naturally. In this sense, the gain comes from applying parent-child binding in a way that is well matched to the trigger-task-step hierarchy of ProAct-75.
> > >
> > > To support this interpretation, we provide two targeted analyses below.
> > >
> > > ### **1. Does Table R6 reflect indiscriminate similarity rather than true parent-child alignment?**
> > >
> > > A natural concern is whether the high cosine similarity in Table R6 simply reflects representational collapse or indiscriminate global similarity. To test this, we randomly permute the parent-child pairing at test time, so that each trigger representation is matched with a non-corresponding task representation, and then recompute cosine similarity.
> > >
> > > **Table R9. Paired and shuffled cosine similarity (full test set).**
> > >
> > > |**Model**|**Cos(t,T)paired**|**Cos(t,T)shuffled**|**Δ**|**Cos(T,s)paired**|**Cos(T,s)shuffled**|**Δ**|
> > > |:-:|:-:|:-:|:-:|:-:|:-:|:-:|
> > > |w/o HBM|0.197|0.188|+0.009|0.406|0.219|+0.187|
> > > |w/ HBM|0.987|0.220|**+0.768**|0.978|0.199|**+0.780**|
> > >
> > > Without HBM, shuffling changes trigger-task cosine only marginally, indicating weak pair-specific alignment. With HBM, the paired-shuffled gap becomes substantially larger at both levels. This shows that the increased similarity is tied to correct hierarchical correspondence, rather than indiscriminate global similarity.
> > >
> > > ### **2. Is the stronger cross-level structure achieved by trivial collapse?**
> > >
> > > To further explain the mechanism, we analyze the geometry of the learned representation space from two complementary angles. **Effective rank (eRank)** measures how broadly information is distributed across the 2048 hidden dimensions [1], while **shared subspace overlap** ($\text{SSO}_{50}$) measures how much task-level variance lies in the principal subspace of trigger representations [2,3]. Concretely, we project task representations onto the top-50 principal components of trigger representations and report the fraction of variance explained.
> > >
> > > **Table R10. Representational geometry under HBM (trigger level).**
> > >
> > > |**Model**|**eRank↑**|**$\text{SSO}_{50}$↑**|
> > > |:-:|:-:|:-:|
> > > |w/o HBM|637|7.7%|
> > > |w/ HBM|934|64.7%|
> > >
> > > Without HBM, cross-level shared structure is weak, as the top-50 principal components of trigger representations explain only 7.7% of task variance. With HBM, shared subspace overlap rises to 64.7%, while effective rank also increases from 637 to 934, indicating stronger cross-level structure without representational collapse.
> > >
> > > ### **Conclusion**
> > >
> > > **Taken together, these two analyses support the interpretation above:** R9 shows stronger alignment on correct parent-child pairs, and R10 shows that this shared cross-level structure is learned without representational collapse.
> > >
> > > **This matches the original motivation of HBM**: the trigger-task-step hierarchy requires explicit cross-level organization, especially under severe long-tail imbalance, while the supervised prediction objective still preserves fine-grained discrimination. HBM therefore contributes a parent-child binding mechanism that is well matched to the hierarchical structure of ProAct-75. As a result, the learned representation becomes partly shared across levels yet still specialized enough to separate child classes.
> > >
> > > This is also consistent with prior self-supervised studies showing that training objectives can reshape representation geometry and effective dimensionality, while avoiding dimensional collapse remains important [4,5].
> > >
> > > ---
> > > ## **Final Appeal**
> > >
> > > **We thank the reviewer again for this helpful follow-up. We will include these analyses in the Appendix to make the representational effect of HBM clearer in the revised version. Given the above clarifications and analyses, we hope this addresses your remaining concern, and we would greatly appreciate reconsideration of the evaluation toward acceptance**
> > >
> > > ---
> > >
> > > ### **References**
> > >
> > > [1] Roy and Vetterli, *The Effective Rank: A Measure of Effective Dimensionality*, 2007, EUSIPCO.
> > >
> > > [2] Raghu et al., *SVCCA: Singular Vector Canonical Correlation Analysis for Deep Learning Dynamics and Interpretability*, 2017, NeurIPS.
> > >
> > > [3] Kornblith et al., *Similarity of Neural Network Representations Revisited*, 2019, ICML.
> > >
> > > [4] Wang et al., *Understanding Contrastive Representation Learning through Alignment and Uniformity on the Hypersphere*, 2020, ICML.
> > >
> > > [5] Jing et al., *Understanding Dimensional Collapse in Contrastive Self-Supervised Learning*, 2022, ICLR.

---

### Official Review · Reviewer_Xfmg · 2026-03-07

**Soundness:** 2
**Presentation:** 2
**Significance:** 2
**Originality:** 3
**Overall Recommendation:** 3
**Confidence:** 4

**Summary:**

The paper introduces a benchmark to study proactive agents that monitor the environment and decide when and how to act. The dataset covers a variety of tasks with multi-level annotations and explicit task graphs that describe dependencies and possible parallel execution. Based on this benchmark, the authors propose ProAct-Helper, a multimodal framework built on an MLLM that performs trigger, task, and step detection and selects proactive actions through task-graph reasoning and entropy-driven heuristic search.

**Compliance With Llm Reviewing Policy:**

Affirmed.

**Final Justification:**

I remain concerned about the paper’s limited scope of proactivity and its reliance on structured task graphs, and will maintain my score.

**Key Questions For Authors:**

- Does training on the proposed benchmark will largely hurt generalization? Has the proposed approach been evaluated on other video reasoning datasets?
- Only three annotators were involved. What was the inter-annotator agreement? Providing more detailed annotation statistics would be helpful.
- Is trigger/proactive evaluation frame-wise or segment-wise? Is there any analysis of under-triggering or over-triggering?
- The use of explicit task graphs may limit scalability, as such structured annotations are costly and rare. Should a well-learned model instead infer the graph internally rather than rely on explicit supervision?

**Limitations:**

yes

**Strengths And Weaknesses:**

**Strength**
- The proposed benchmark emphasizes proactive behavior and combines perception with task-graph reasoning, enabling systematic evaluation of different aspects of proactive agents.
- The paper introduces an MLLM-based framework that supports multi-level state perception and action search using task graphs.
- The paper includes thorough ablation studies, and experiments show improvements across multiple tasks compared with strong closed-source models.

**Weakness**
- The benchmark offers limited novelty, as it closely resembles existing Online action detection and anticipation datasets. The claimed distinction of “proactive” behavior is weak, since simply detecting non-background actions already captures much of the intended behavior.
- The benchmark is mainly limited to procedural tasks and does not reflect the complexity of real-world scenarios. In practice, tasks often lack clear hierarchies or dependencies and involve overlapping or multi-label actions, making detection and task-graph construction far more challenging.
- The work is framed as proactive agent research, but the proposed framework mainly uses SFT-trained MLLMs with heuristic task-graph reasoning rather than learning  from the environment.  The “agent” formulation appears somewhat overstated.
- The framework offers limited methodological novelty, relying mainly on a standard contrastive loss. It is specifically designed for the benchmark and may struggle to generalize to unseen instructions or domains, restricting its ability to exhibit true proactive behavior.

---

> ### Author Rebuttal · Authors · 2026-03-29
>
> ### W1: Benchmark novelty
> ProAct-75 targets proactive response, where an agent monitors the scene and determines when and how to intervene. In human-robot collaborative settings, inferring intentions from observations is a prerequisite for intervention [1] (Section 2.1). Our task, step, and future prediction tasks serve this perceptual role, and their overlap with detection is by design.
>
> **Two tasks have no detection counterpart.** Trigger detection asks whether intervention is needed now, not what action is happening. We annotate each step with a trigger description, and Table R5 further shows that models exhibit opposite failure modes. Action selection asks which action complements the human under graph constraints. Prior methods tend to mirror the human's next step [2], which can block parallel workflows (Figure 1). In contrast, our method is designed to favor parallel threads, while closed-source LLMs generally show weaker preference for parallel execution despite task-graph guidance (Section 6.5).
> ### W2: Limited to procedural tasks
> Our benchmark already captures the complexity raised. ProAct-75 covers **three domains** (Section 4.1) across kitchens, bedrooms, workshops, offices, outdoor scenes, and more. On hierarchies, **88% of graphs are multi-level with 16.9% mid-level nodes** that encapsulate reusable sub-procedures for modular maintenance. On graph construction, our **semi-automated pipeline (Section 4.2)** has LLMs propose temporal precedences and human experts verify, making it scalable beyond manual annotation. While 24.1% of actions are shared across tasks, this overlap is natural and ambiguity can be reduced with longer observation context.
> ### W3: "Agent" formulation
> Our framework integrates perception and executable action selection for proactive response. We agree that the system relies on SFT with heuristic supervision rather than direct environment interaction, and we position it as a pragmatic intermediate step toward a fully learned agent. Applying RL from scratch is challenging here because the problem is inherently multi-objective, and finding a Pareto-optimal trade-off across the four metrics is non-trivial. In addition, SS is an episode-level delayed reward, while the human’s reactive behavior (Section 6.4) makes the environment non-stationary. Our heuristic finds high-quality solutions (Table 3, SS=9.87, 60% above the best LLM), which then serve as SFT labels and provide a practical starting point for future learned planning with structured supervision.
> ### W4: Methodological novelty
> HBM borrows the contrastive formulation but does not directly apply standard contrastive learning. The trigger-task-step hierarchy has severe class imbalance (2 trigger classes, 75 tasks, 1,251 steps), and autoregressive supervision alone cannot enforce cross-level consistency. HBM binds parent-child representations to improve rare-class separability. Ablation (Table 2) and representation analysis (Table R6; see our response to Reviewer Hyas-W1 & Q3) confirm its effectiveness.
> ### Q1: Generalization.
> We conduct three OOD experiments (cross-view, actor-split, dataset-split) detailed in our response to Reviewer YwTj-W1, where trigger F1 drops by only 3.23%, suggesting relatively robust generalization.
> ### Q2: Inter-annotator agreement
> Three annotators worked over six weeks following a divide-then-review protocol (Section 4.1). Round 1 was self-review, Round 2 was cross-review by other experts. Low Round 2 rates confirm quality converged after self-review.
>
> **Table R4. Annotation correction rates.**
> | Source | R1 (video %) | R2 (step %) |
> |--------|:---:|:---:|
> | UCF-Crime | 24.7 | 0.2 |
> | COIN | 82.7 | 13.9 |
> | Ours | 75.3 | 5.2 |
> ### Q3: Trigger evaluation
> Evaluation is **frame-wise** (per-keyframe). We provide a new under/over-triggering analysis with False Positive Rate (FPR, over-triggering) and False Negative Rate (FNR, under-triggering).
>
> **Table R5 Under-triggering and over-triggering analysis.**
> | Method | FPR↓| FNR↓ | mF1↑ |
> |--------|------|------|------|
> | GPT-4o | 39.2 | 59.2 | 46.7 |
> | Gemini-2.5-Pro | 77.4 | 9.2 | 55.9 |
> | ProAct-Helper (7B) | 34.5 | 13.2 | 76.6|
>
> Models exhibit opposite failure modes, where GPT-4o under-triggers while Gemini over-triggers. Only ProAct-Helper controls both errors, achieving the best mF1 of 76.6%. This highlights that **trigger detection is not reducible to action recognition**, as the core question is “when to act”. We will include this analysis in Appendix.
> ### Q4: Graph supervision and scalability
> Please see our response to Reviewer Hyas-W2, Q4 & L2. We provide a new end-to-end variant showing that planning can be learned without heuristic inference, while our benchmark’s graphs serve as scalable supervision for future graph-inference methods.
> ### References
> [1] Baker et al., Action Understanding as Inverse Planning, 2009, Cognition.
>
> [2] Koppula et al., Anticipating Human Activities Using Object Affordances for Reactive Robotic Response, 2015, T-PAMI.

---

> > ### Author Rebuttal · Reviewer_Xfmg · 2026-04-02
> >
> > I appreciate the authors’ rebuttal and clarifications. However, I remain unconvinced about the benchmark’s significance and contribution. Despite being framed as proactive, it relies on predefined hierarchical labels, limiting generalization and falling short of open-ended multimodal settings. The “agent” aspect also feels overstated, and the design appears to be a modest extensions of action detection. Therefore, I keep my score unchanged.

---

> > > ### Author Response · Authors · 2026-04-02
> > >
> > > We appreciate the reviewer’s time and effort. Some key points in our previous response may not have been sufficiently clear, which may have caused misunderstanding about our design rationale and implementation details. We therefore clarify them below.
> > >
> > > ### **1. Benchmark significance beyond action detection**
> > >
> > > Existing video-based proactive settings often focus on action detection and anticipation [2], while leaving two related questions less explicit in evaluation. **One is whether intervention is needed in the current context. The other is what response should follow once the task state is recognized.** Our benchmark makes these two questions explicit through trigger prediction and proactive action selection.
> > >
> > > In our annotations, not all observable events are meaningful triggers. For example, actions such as walking or drinking water may be observable but do not necessarily require intervention. **Trigger prediction therefore cannot always be reduced to recognizing what action is happening.** Instead, it depends on whether intervention is needed in the current context, as reflected in our trigger descriptions. **It also serves as a practical gate, as non-trigger cases avoid downstream decoding and reduce average decoded tokens by 17.4%.**
> > >
> > > More importantly, **prediction alone is not the end of proactive response**. Once intervention is needed, the system must still decide how to act. This is why we include proactive action selection, which connects perception to executable response.
> > >
> > >
> > > ### **2. Structured design and generalization**
> > >
> > > Proactive response is inherently structured, as it connects higher-level objectives with executable steps. Thus, hierarchical supervision is a natural formalization of when and how to intervene. In our understanding, the generalization concern more directly relates to explicit task structure for action selection.
> > >
> > > At the same time, explicit task structure is also common in procedural video benchmarks. For example, **Ego-Exo4D explicitly annotates per-activity task graphs and evaluates task-graph-related reasoning in its Keystep challenge [1]**. Our benchmark follows the same principle for proactive response, while extending the graph with mid-level nodes and AND/OR conditions (Section 3.1).
> > >
> > > We understand the reviewer’s concern about generalization and the gap to open-ended settings. To address this, we further provided OOD results (YwTj-W1) and task-graph noise robustness results (Hyas-Q1&Q2) in rebuttal. These results suggest that the current structured formulation does not prevent broader generalization. Rather, it can serve as a structured warm start for future policy learning, while more open-ended settings remain a natural next step.
> > >
> > > ### **3. The grounding of “proactive” and “agent”**
> > >
> > > We also clarify our use of the term **proactive agent**. As stated in the Introduction, proactive agents align with higher-level objectives and act by determining when and how to respond. This is the sense in which we use the term **proactive**. In our benchmark, these higher-level objectives are instantiated as assistance, maintenance, and safety monitoring, with task completion as the concrete goal.
> > >
> > > Regarding the term **agent**, Russell et al. define it in *Artificial Intelligence: A Modern Approach* as follows [3]:
> > >
> > > > “An agent is anything that can be viewed as perceiving its environment through sensors and acting upon that environment through effectors.”
> > >
> > > More importantly, **we respectfully refer the reviewer to our experiment in Section 6.4.** In that experiment, the human and robot are simulated from the initial state. The human follows the annotated trajectory unless preempted by the robot, then switches to a parallel branch, while the robot takes predicted human actions as input and selects from the legal set using an entropy-based heuristic policy. **Their actions jointly drive task-state transitions, and the new state then determines the next observation and action-selection input.**
> > >
> > > In this sense, the term “agent” here is operationally grounded, rather than merely a label for static prediction. A simulation is illustrated in Figure 8 ([a](https://github.com/only4anonymous/ProAct-Helper/blob/main/figure4rebuttal/trash_bin_1.png), [b](https://github.com/only4anonymous/ProAct-Helper/blob/main/figure4rebuttal/assemble_sofa_1.png); click to view).
> > >
> > > ---
> > > ## **Final Appeal**
> > >
> > > **We again thank the reviewer for the helpful feedback. We hope these clarifications better reflect the scope and contribution of our work. If the reviewer finds these concerns sufficiently addressed, we would greatly appreciate reconsideration toward acceptance.**
> > >
> > > ---
> > >
> > > ### **References**
> > >
> > > [1] Grauman et al., *Ego-Exo4D: Understanding Skilled Human Activity from First- and Third-Person Perspectives*, 2024, CVPR.
> > >
> > > [2] Bhagat et al., Let Me Help You! Neuro-Symbolic Short-Context Action Anticipation, 2024, RA-L.
> > >
> > > [3] Russell and Norvig, Artificial Intelligence: A Modern Approach, 1995, Prentice Hall.

---

### Official Review · Reviewer_2o5s · 2026-03-09

**Soundness:** 3
**Presentation:** 3
**Significance:** 3
**Originality:** 3
**Overall Recommendation:** 4
**Confidence:** 3

**Summary:**

The paper introduces ProAct-75, a new benchmark designed to train and evaluate proactive robotic agents that do not just follow instructions but take initiative based on environmental monitoring. The dataset spans 75 tasks across three scenarios: assistance, maintenance, and safety monitoring, featuring over 91,000 step-level annotations. A key contribution is the inclusion of explicit task graphs (Directed Acyclic Graphs with AND/OR dependencies) that encode step precedence and parallel execution possibilities.

The authors also propose ProAct-Helper, an MLLM-based framework (built on Qwen2.5-VL) that integrates a Hierarchical Binding Module (HBM) to ensure semantic consistency across hierarchical detection tasks (trigger, task, and step). To select the best proactive action, the model uses an entropy-driven heuristic search that leverages the task graph to prioritize parallel execution threads, allowing the robot to work independently rather than merely mirroring the human's next step. Experiments demonstrate that ProAct-Helper outperforms strong baselines in trigger detection, saved steps, and parallel action rates.

**Compliance With Llm Reviewing Policy:**

Affirmed.

**Final Justification:**

Addressed my questions on generalization and latency. I will go with my current positive assessment, while the novelty and benchmark contribution needs to be strengthened.

**Key Questions For Authors:**

1.	Zero-Shot Transfer: How does the ProAct-Helper framework perform on entirely unseen tasks? Since the action selection relies on the task graph, would the agent be able to function if it had to infer the graph structure from a few-shot demonstration or a textual description?
2.	Latency and Real-time Feasibility: The pipeline involves an MLLM with HBM followed by a heuristic search. What is the typical end-to-end latency for a single decision point, and is this currently suitable for real-time closed-loop robotic control?

**Limitations:**

The authors have adequately discussed several limitations, failure cases and ethical considerations. They address privacy concerns by blurring faces and sensitive information in self-collected data. They also acknowledge a technical limitation where strict task-graph feasibility can stall rollouts if human behavior doesn't perfectly match the annotated graph, necessitating a deadlock-prevention alignment during simulation. However, a more detailed discussion on the potential risks of autonomous triggering in the safety monitoring (UCF-Crime) scenario would further strengthen the paper as false positives could lead to unnecessary interventions in this scenario.

**Strengths And Weaknesses:**

•	Strengths:
1. The experimental design is rigorous, and uses a diverse set of source data (Ego-Exo4D, COIN, UCF-Crime) and self-collected videos to ensure coverage. The evaluation metrics fit the proactive response problem. The use of Saved Steps (SS) and Parallel Action (PA) are indicative of the advantages of proactive agents over passive agents.
2. The paper is well-structured. The formalization of the task graph using AND/OR semantics is clear and mathematically grounded. Figures 1 and 2 effectively illustrate the difference between passive and proactive responses through concrete examples like trash-bin replacement. Figure 4 gives a good summary of the Helper framework.
3. This work addresses the reliance on sequential, passive instruction following in human-robot collaboration. By focusing of building proactive skills, this work tries to reduce repetitive instructions by humans and give the robots a bit more autonomy. By providing a structured way to handle parallel workflows, it enables robots to significantly reduce human cognitive load and total task time.
4. The integration of explicit, DAG-based task graphs into a vision-based benchmark is a novel and valuable contribution to the field. The entropy-driven heuristic for action selection is a creative way to balance "following the human" with "executing independent threads".

•	Weaknesses:
1. The action selection mechanism is highly dependent on the accuracy of upstream perception. While the HBM improves perception, the authors note that hallucinated actions or incorrect step detection can lead to "forced waits" or illegal actions in the simulation.
2. The implementation details for the text-only simulation (e.g., the deadlock-prevention safeguard) are relegated to the appendix, even though they are critical for understanding how the model handles the strict constraints of the task graphs.

---

> ### Author Rebuttal · Authors · 2026-03-29
>
> ### W1: Upstream perception dependency
> We agree this dependency is inherent to any perception-then-planning pipeline, and we deliberately designed safeguards for it. At the planning level, the intersection filter A_cand = A_legal ∩ A_pred (Section 6.1) guarantees that every selected action is executable. If the entire set of predicted actions is invalid, the system defaults to a forced wait as a safe fallback rather than taking an illegal action. At the perception phase, HBM reduces upstream errors by grounding predictions in cross-hierarchy context (Table 2). Our trigger analysis (Table R5; see our response to Reviewer Xfmg-Q3) further shows ProAct-Helper achieves the highest trigger mF1 of 76.6%, reducing one major source of downstream planning errors.
>
> ### W2: Simulation details in Appendix
> Thank you for this suggestion. Due to the page limit, we placed the simulation implementation details in Appendix A. We will move these to the main text in the revision to improve readability.
>
> ### Q1: Zero-shot transfer.
> We address this from two aspects. **For unseen tasks**, the action selection module itself is task-agnostic. Given a valid task graph, it can operate on unseen tasks without retraining. The main challenge is therefore perception. To assess this, we conduct three OOD experiments (cross-view, actor-split, and dataset-split; details in our response to Reviewer YwTj-W1). In particular, under the dataset-split setting where training and testing have completely disjoint task categories, trigger F1 drops by only 3.23%, suggesting that trigger-level proactive cues transfer reasonably well beyond seen tasks.
>
> **For graph acquisition**, our current system assumes the task graph is provided as input. In practice, task graphs for daily activities can be constructed from instructional videos, how-to articles, or recipe databases, and our semi-automated pipeline (Section 4.2) already demonstrates this feasibility. More generally, we agree that automatically inferring task graphs from few-shot demonstrations or textual descriptions is an important future direction. Our benchmark provides ground-truth task graphs (Section 4.2) and natural-language context cues, including scenario and trigger descriptions (Section 4.1), which can support both training and evaluation for this line of research.
> ### Q2: Latency and real-time feasibility.
> We report detailed wall-clock latency in Appendix B.3 (Table 5). ProAct-Helper-7B achieves **2.75s perception + 0.08s action selection = 2.83s per decision point**. Open-source 30B+ baselines require 15-46s because our evaluation follows a two-stage protocol (Section 6.1) that feeds the full task graph as context (with additional structured guidance such as AND/OR constraints, thread definitions, and action-selection rules), so the model can map its free-form predictions to the correct graph node, resulting in longer inputs. Their larger parameter counts and longer output sequences (lacking the special tokens we trained for compact structured decoding) further increase latency.
>
> Proactive response operates at the action selection stage, which is later instantiated by low-level motion planners into executable moves. In collaborative tasks, **human actions typically last 5-15 seconds per step**, so 2.83s fits comfortably within this decision window. Reducing perception latency further through model distillation or inference acceleration is a worthwhile direction for future work.
>
> ### L1: Safety monitoring false positives.
> Thank you for this important point. Our trigger error analysis (Table R5, see our response to Reviewer Xfmg-Q3) quantifies this risk. ProAct-Helper-7B achieves FPR=34.5%, substantially lower than Gemini 2.5 Pro's 77.4% but not yet negligible. For safety-critical deployments such as UCF-Crime monitoring, we recommend two mitigation strategies. First, confidence-based suppression, where triggers below a calibrated threshold are withheld before intervention. Second, a human-in-the-loop confirmation step before executing high-stakes interventions, consistent with the "safe interruptibility" framework proposed by Hadfield-Menell et al. [1]. We will add a dedicated discussion of these risks and mitigations in the revision.
>
> ### References
> [1] Hadfield-Menell et al., The Off-Switch Game, 2017, IJCAI.

---

> > ### Author Rebuttal · Reviewer_2o5s · 2026-04-02
> >
> > Authors' response has resolved my comments. I will retain my positive assessment and keep my score.

---

### Official Review · Reviewer_YwTj · 2026-03-19

**Soundness:** 4
**Presentation:** 4
**Significance:** 3
**Originality:** 3
**Overall Recommendation:** 5
**Confidence:** 4

**Summary:**

The paper studies proactive agents which know high-level objectives (task), when and how to intervene. To this end, a new dataset called ProAct-75 -- spanning assistance, maintenance, and safety monitoring, and including explicit task graphs (encoding dependencies between steps/actions and parallel execution possibilities) -- was introduced. Furthermore, a baseline multimodal LLM named ProAct-Helper (with cross-level (trigger, step, task) alignment and proactive action selection) was proposed.

**Compliance With Llm Reviewing Policy:**

Affirmed.

**Key Questions For Authors:**

1. In Fig 4, H is an input to the pooling, whereas in the text H is the output of pooling. Please resolve this discrepancy.
2. After you produce H vectors and apply contrastive loss, do you apply an MLP projection to H vectors? Forcing the raw H vectors for different levels seems like a very strong objective.
3. Compared to LLM inference cost, how large is the cost of proactive action selection (including entropy-driven heuristic search)?

Minor
1. in page 1: "Continuously observations" -> Continuously observing

**Limitations:**

yes.

**Strengths And Weaknesses:**

Strengths
1. The combination of proactive assistance and task graph (dependency, parallelism) is novel and promising idea.
2. The writing (including mathematical notations) is rigorous.
3. The benchmark still seems challenging to SOTA LLMs, hence it can be a valuable benchmark to the field.
4. Related Work section was a good read.

Weaknesses
1. I think authors could have tried testing out-of-distribution (e.g. outside 75 tasks) performance.

---

> ### Author Rebuttal · Authors · 2026-03-29
>
> ### 1. W1 OOD performance.
> Following OOD protocols in activity recognition [2][3], we test generalization under three distribution shifts. The results and analysis are as follows.
>
> **(1) Cross-view OOD.** We report cross-view OOD results in Appendix B.2 (Table 4), where models trained on best-view videos are evaluated on other views with different viewing angles and heavier occlusion, reflecting a common challenge in multi-view recognition.
>
> **(2) Actor-split OOD.** On our self-collected videos, we train ProAct-Helper (Qwen2.5-VL-3B-Instruct) on Actors 1-4 and evaluate on held-out Actor 5, with the control training on all five actors.
>
> **Table R1. Actor-split OOD.**
> | Setting | Trig. F1 | Task F1 | Step F1 | Future ED↓ |
> |---------|----------|---------|---------|------------|
> | OOD | 86.46 | 64.21 | 34.97 | 2.670 |
> | Full | 93.00 | 94.64 | 89.97 | 0.563 |
>
> Performance degradation mainly concentrates at the fine-grained step level, which is consistent with prior cross-subject findings in activity recognition [2]. Variations in execution style, temporal rhythm, and interaction details across actors make task and step recognition substantially harder, and these errors further propagate to future prediction. This also suggests that improving generalization to unseen actors remains an important direction for future work.
>
> **(3) Dataset-split OOD.** We train on public datasets only (Ego-Exo4D, COIN, and UCF-Crime) and test on our self-collected videos, whose task categories are disjoint from training. The control uses the full training set in Table 1, and both settings are trained for 10 epochs. Since task labels are disjoint and step labels are not directly aligned, we report trigger detection only.
>
> **Table R2. Dataset-split OOD.**
> | Setting | Trig. mAcc | Trig. mF1 | Trig. Acc | Trig. F1 |
> |---------|-----------|----------|----------|---------|
> | OOD | 53.71 | 66.82 | 79.13 | 87.03 |
> | Full | 70.80 | 82.38 | 85.90 | 90.26 |
>
> **Trigger F1 drops only modestly from 90.26 to 87.03**, suggesting relatively robust trigger recognition under dataset-level shift. The larger drop in trig mF1 indicates uneven transfer across trigger types, likely due to different degrees of visual overlap with the training data. This also highlights that cross-task generalization in in-the-wild settings remains an important direction for future research.
> ### Q1 & Minor: Figure 4 notation and typo.
> Thank you for catching this. In our notation, h denotes individual token embeddings for trigger, task, and step, while H denotes the mean-pooled vectors. We will correct both the notation and the typo in the revision.
> ### Q2: MLP projection head.
> Thank you for this insightful question. We agree that exploring better representation training strategies is important. We provide both our design rationale and new ablation results.
>
> Following your suggestion, we trained two variants with a 2-layer projection head of the form Linear(H, d) → ReLU → Linear(d, d), where H=2048 is the Qwen2.5-VL hidden size and d∈{128, 256}, following the design in [1]. For rapid validation, we use the same 1/8 stratified mini-split as our ablation study (Section 6.3). All other hyperparameters are identical.
>
> **Table R3. MLP projection head ablation.**
> | Method | Trig. mF1 | Task mF1 | Step mF1 | ED↓ | Val Loss↓ |
> |--------|----------|----------|----------|-----|----------|
> | w/o MLP (Ours) | **70.18** | **23.10** | 7.18 | **3.94** | **0.306** |
> | w/ MLP (d=128) | 67.55 | 21.41 | 6.99 | 4.05 | 0.649 |
> | w/ MLP (d=256) | 68.60 | 21.18 | **7.25** | 4.03 | 0.694 |
>
> **The MLP variants underperform across all metrics, with validation loss nearly 2x higher**, indicating worse generalization. A likely reason is that the binding loss is jointly optimized with the autoregressive generation objective, while the randomly initialized MLP adds extra transformations that make this alignment harder to learn. This is consistent with our training logs, where the MLP variants maintain **7–16x higher binding losses** throughout training. Overall, the results suggest that directly applying the loss on H better preserves hierarchical alignment for generation than simply adding an MLP, motivating further exploration of better feature representations for cross-level binding. We will include this ablation in the Appendix.
> ### Q3: Cost of action selection.
> We report both perception and planning costs in Appendix B.3 (Table 5). The action selection step alone takes **0.08s** per step on average, which is faster than even the lightest MLLM performing the same task (Qwen2.5-VL-3B at 0.16s). The total pipeline latency is dominated by the perception module rather than action selection.
> ### References
> [1] Chen et al., A Simple Framework for Contrastive Learning of Visual Representations, 2020, ICML.
>
> [2] Das et al., Toyota Smarthome: Real-World Activities of Daily Living, 2019, ICCV.
>
> [3] Liu et al., NTU RGB+D 120: A Large-Scale Benchmark for 3D Human Activity Understanding, 2020, IEEE TPAMI.

---

> > ### Author Rebuttal · Reviewer_YwTj · 2026-04-06
> >
> > Thank you for your rebuttal. My questions are all addressed and I do not have any more questions.

---

### Decision · Program_Chairs · 2026-04-30

**Decision:**

Accept (regular)

**Comment:**

The reviewers generally agree that the benchmark is well constructed and addresses a meaningful gap. The main criticism is whether the benchmark and method are sufficiently novel versus being an engineering-driven extension of action detection (Xfmg, Hyas). The "agent" framing is somewhat overclaimed given the reliance on SFT with heuristic supervision rather than learned environment interaction, though the authors acknowledge this as a pragmatic intermediate step.

The authors provided a strong rebuttal with substantial new experiments. While the AC agrees that the "agent" framing is somewhat overclaimed and the scope is limited to structured procedural tasks, the benchmark does fill a gap by making trigger detection and proactive action selection explicit evaluation targets. Overall, the contributions are sufficient for acceptance.